# Direct Evolutionary Optimization of Variational Autoencoders With Binary Latents

## Abstract

Many types of data are generated at least partly by discrete causes that are sparsely active. To model such data, we here investigate a deep generative model in the form of a variational autoencoder (VAE) which can learn a sparse, binary code for its latents. Because of the latents' discrete nature, standard VAE training is not possible. The goal of previous approaches has therefore been to amend (i.e., typically anneal) discrete priors in order to train discrete VAEs analogously to conventional ones. Here, we divert much more strongly from conventional VAE training: We ask if it is also possible to keep the discrete nature of the latents fully intact by applying a direct, discrete optimization for the encoding model. In doing so, we (1) sidestep standard VAE mechanisms such as sampling approximation, reparameterization trick and amortization, and (2) observe a much sparser encoding compared to autoencoders that use annealed discrete latents. Direct optimization of VAEs is enabled by an evolutionary algorithm in conjunction with truncated posteriors as variational distributions, i.e. by a combination of methods which is here for the first time applied to a deep model. We first show how the discrete variational method (A) ties into gradient ascent for network weights, and how it (B) uses the decoder network to select binary latent states for training. Sparse codes have prominently been applied to image patches, where latents encode edge-like structure. For our VAEs, we maintain this prototypical application domain and observe the emergence of much sparser codes compared to more conventional VAEs. To allow for a broad comparison to other approaches, the emerging encoding was evaluated on denoising and inpainting tasks, which are canonical benchmarks for image patch models. For datasets with many, large images of single objects (ImageNet, CIFAR etc) deep generative models with dense codes seem preferable. For image patches, however, we observed advantages of sparse codes that give rise to state-of-the-art performance in 'zero-shot' denoising and inpainting benchmarks. Sparse codes can consequently make VAEs competitive on tasks where they have previously been outperformed by non-generative approaches.

## 1 Introduction & Related Work

Objects or edges in images are either present or absent, which suggests the use of discrete latents for their representation. There are also typically only few objects per image (of all possible objects) or only few edges in any given image patch (of all possible edges), which suggests a sparse code (e.g. Olshausen & Field, 1996; Titsias & Lázaro-Gredilla, 2011; Goodfellow et al., 2013; Sheikh et al., 2014). In order to model such and similar data, we study a novel, direct optimization approach for variational autoencoders (VAEs), which can learn a sparse, discrete encoding. VAEs (Kingma & Welling, 2014; Rezende et al., 2014) in their many different variations, have successfully been applied a large number of tasks including semi-supervised learning (e.g. Maaløe et al., 2016), anomaly detection (e.g. Kiran et al., 2018), sentence interpolation (Bowman et al., 2016) or music interpolation (Roberts et al., 2018) to name a few. The success of VAEs, in these tasks, rests on a series of methods that enable the derivation of scalable training algorithms to optimize VAE parameters. These methods were originally developed for Gaussian priors; to account for VAEs with discrete latents, novel methodology had to be introduced (we elaborate below and later in App. A).

The training objective of VAEs is derived from a likelihood objective, i.e., we seek model parameters $\Theta$ of a VAE that maximize the data log-likelihood, $L(\Theta) = \sum_n \log \left( p_\Theta(\vec{x}^{(n)}) \right)$, where we denote

by $\vec{x}^{(1:N)}$ a set of $N$ observed data points, and where $p_\Theta(\vec{x})$ denotes the modeled data distribution. Like conventional autoencoders (e.g., Bengio et al., 2007), VAEs use a deep neural network (DNN) to generate (or decode) observables $\vec{x}$ from a latent code $\vec{z}$. Unlike conventional autoencoders, however, the generation of data $\vec{x}$ is not deterministic but it takes the form of a probabilistic generative model. For VAEs with binary latents, we here consider the following generative model:

$$p_\Theta(\vec{z}) = \text{Bern}(\vec{z}; \vec{\pi}) = \prod_h \left( \pi_h^{z_h} (1 - \pi_h)^{(1-z_h)} \right), \quad p_\Theta(\vec{x} \mid \vec{z}) = \mathcal{N}\left(\vec{x}; \vec{\mu}(\vec{z}; W), \sigma^2 \mathbb{I}\right), \quad (1)$$

with $\vec{z} \in \{0, 1\}^H$ being a binary code parameterized by priors $\vec{\pi}$, $\vec{x} \in \mathbb{R}^D$, and the non-linear function $\vec{\mu}(\vec{z}; W)$ being a DNN that outputs the mean of the Gaussian distribution. $p_\Theta(\vec{x} \mid \vec{z})$ is commonly referred to as *decoder*. The set of model parameters is $\Theta = \{\vec{\pi}, W, \sigma^2\}$, where $W$ incorporates DNN weights and biases. We assume homoscedasticity of the Gaussian distribution, but note that there is no obstacle to generalizing the model by inserting a DNN non-linearity that outputs a correlation matrix. Similarly, the algorithm could easily be generalized to different noise distributions should the task at hand call for it. For the purpose of this work, however, we will focus on the elementary VAEs given by Eqn. (1).

For conventional and discrete VAEs, essentially all optimization approaches seek to approximately maximize the log-likelihood using the following series of methods (we elaborate in App. A):

(A) Instead of the log-likelihood, a variational lower-bound (a.k.a. ELBO) is optimized.

(B) VAE posteriors are approximated by an *encoding model*, i.e. by a specific distribution (usually Gaussian) parameterized by one or more DNNs.

(C) The variational parameters of the encoder are optimized using gradient ascent on the lower bound, where the gradient is evaluated based on sampling and the reparameterization trick to obtain sufficiently low-variance and yet efficiently computable estimates.

(D) Using samples from the encoder, the parameters of the decoder are optimized using gradient ascent on the variational lower bound.

Optimization procedures for VAEs with discrete latents follow the same steps (Points A to D). However, discrete or binary latents pose substantial further obstacles for learning, mainly due to the fact that backpropagation through discrete variables is generally not possible or biased (Rolfe, 2016; Bengio et al., 2013). Key elements of gradient estimators commonly applied for discrete latent VAEs include, e.g., reparameterizations of continuous approximations to discrete latents (Jang et al., 2016) or score function based reformulations yielding a surrogate loss (Schulman et al., 2015, also see Related Work and App. B.6). While accomplishing, in different senses, the goal of maintaining standard VAE training as developed for continuous latents (i.e., learning procedures and/or learning objectives that allow for gradient-based learning of the encoder and decoder DNNs), gradient estimation methods apply significant amounts of methodology *additional* to the learning methods conventionally applied for VAE optimization. These additional methods (compare Fig. 7), their accompanying design decisions and used hyper-parameters (e.g. parameters for annealing to 'soften' discrete distributions) do increase the complexity of the system that has to be optimized. Furthermore, the additional methods usually impact the learned representations. For instance, softening of discrete distributions, e.g. by using 'Gumbel-softmax' (Jang et al., 2016) or 'tanh' approximations (Fajtl et al., 2020) seems to favor dense codes. While dense codes (as also used by conventional VAEs and GANs) can result in competitive performance for a subset of the above discussed tasks, other recent contributions point out advantages of sparse codes, e.g., in terms of disentanglement (Tonolini et al., 2020) or robustness (Sulam et al., 2020; Paiton et al., 2020).

In order to avoid adding methods for discrete latents to those already in place of standard VAEs, it may be reasonable to investigate more direct optimization procedures that do not require, e.g., a softening of discrete distributions or other mechanisms. Such a direct approach is challenging, however, because once DNNs are used to define the encoding model (as commonly done), we require methodologies for discrete latents to estimate gradients for the encoder (as done via sampling and reparameterization; see Points C and D). A direct optimization procedure, as we investigate here, consequently has to change VAE training substantially. For the data model (1), we will maintain the variational setting (Point A) and a decoding model with DNNs as non-linearity. However, we will not use an encoder model parameterized by DNNs (Point B). Instead, the variational bound will be increased w.r.t. an implicitly defined encoder model which allows for an efficient discrete optimization. The procedure does not require gradients to be computed for the encoder such that discrete latents are addressed without the use of reparameterization trick and sampling approximations.

**Related Work.** In order to maintain the general VAE framework for encoder optimization in the case of discrete latents, different groups have suggested different possible solutions: work by Rolfe (2016), for instance, extends VAEs with discrete latents by auxiliary continuous latents such that gradients can still be computed. Work on the concrete distribution (Maddison et al., 2016) or Gumbel-softmax distribution (Jang et al., 2016) proposes newly defined continuous distributions that contain discrete distributions as limit cases. Work by Lorberbom et al. (2019) merges the Gumbel-Max reparameterization with the use of direct loss minimization for gradient estimation, enabling efficient training on structured latent spaces (for further improved Gumbel-softmax versions also see Kool et al., 2020; Potapczynski et al., 2020; Paulus et al., 2021). Furthermore, work, e.g., by van den Oord et al. (2017), and Roy et al. (2018) combines VAEs with a vector quantization (VQ) stage in the latent layer. Latents become discrete through quantization but gradients for learning are adapted from latent values before they are processed by the VQ stage. Similarly, Tomczak & Welling (2018) use what they call (learnable) pseudo-inputs which determine a mixture distribution as prior, and the ELBO then contains an additional regularization for consistency between prior and average posterior. Tonolini et al. (2020) extend this work and introduce an additional DNN classifier. The classifier selects pseudo-inputs and the classifier weights are learned instead of the pseudo-inputs themselves. Using their approach, Tonolini et al. (2020) also argue for the benefits not only of discrete latents but of a sparse encoding in the latent layer in general. Fajtl et al. (2020) base their approach on a deterministic autoencoder and use a tanh-approximation of binary latents and a projections to spheres in order to treat binary latents. Targeting not only discrete VAE optimization but more general approaches such as probabilistic programming or general stochastic automatic differentiation, Bingham et al. (2019); van Krieken et al. (2021) apply gradient estimators for discrete random variables which optimize surrogate losses (Schulman et al., 2015) derived based on the score function (Foerster et al., 2018) or other methods (van Krieken et al., 2021). For the derivation of efficient, unbiased any-order derivatives, techniques for variance reduction of estimators are identified as key components. van Krieken et al. (2021) mention exponential scaling behavior which, however, can be addressed using efficient parallelized computations. Furthermore related are methodologies for gradient estimation for unknown ('black-box') functions of discrete or continuous random variables as studied by (Grathwohl et al., 2018), who optimize neural network-based surrogate losses.

In terms of numerical evaluation, the above discussed approaches demonstrate the benefits of the respective new methodology on very different experiments. For instance, van den Oord et al. (2017) focus on image, video and speech generation capabilities of their approach, and report concrete values e.g. for phoneme classification for their evaluation. Roy et al. (2018) show improved image generation capabilities, e.g., on CIFAR and report accuracy scores for machine translation similar to autoregressive baselines. Tomczak & Welling (2018) show competitive likelihood results in the unsupervised permutation invariant setting for many data sets; and Fajtl et al. (2020) evaluate their approach based on accuracy/sensitivity tradeoffs among other comparisons. Tonolini et al. (2020) focus on the competitive capabilities of their VAEs to learn meaningful (disentangled) features; and Lorberbom et al. (2019) focus on the benefits of structured priors. Numerical evaluations of their VAEs and different numbers of categoric latents show better performance of their direct loss minimization than Gumbel-softmax VAEs, and the same applies for comparisons in terms of semi-supervised learning. Similarly to Tonolini et al. (2020), disentanglement results for the relatively large CelebA dataset are also shown by Lorberbom et al. (2019). For this and many other datasets, large DNNs are required while the latent layers are often kept relatively small.

## 2 Direct Variational Optimization

Let us consider the variational lower bound of the likelihood. If we denote by $q_\Phi^{(n)}(\vec{z})$ the variational distributions with parameters $\Phi = (\Phi^{(1)}, \ldots, \Phi^{(N)})$, then the lower bound is:

$$\mathcal{F}(\Phi, \Theta) = \sum_n \mathbb{E}_{q_\Phi^{(n)}} \big[ \log \big( p_\Theta(\vec{x}^{(n)} \,|\, \vec{z}) \, p_\Theta(\vec{z}) \big) \big] - \sum_n \mathbb{E}_{q_\Phi^{(n)}} \big[ \log \big( q_\Phi^{(n)}(\vec{z}) \big) \big], \qquad (2)$$

where we sum over all data points $\vec{x}^{(1)}, \ldots, \vec{x}^{(N)}$, and where $\mathbb{E}_{q_\Phi^{(n)}} \big[ h(\vec{z}) \big]$ denotes the expectation value of a function $h(\vec{z})$ w.r.t. distribution $q_\Phi^{(n)}(\vec{z})$. The general challenge for the maximization of $\mathcal{F}(\Phi, \Theta)$ is the optimization of the encoding model $q_\Phi^{(n)}$. VAEs with discrete latents add to this challenge the problem of taking gradients w.r.t. discrete latents. Seeking to avoid such derivatives, we, for our purposes, do *not* use a DNN for the encoding model. Consequently, we need to define an

*alternative* encoding model $q_\Phi^{(n)}$, which has to remain sufficiently efficient. Considering prior work on generative models with discrete latents, variational distributions based on truncated posteriors offer themselves as such an alternative (Lücke & Sahani, 2008). Truncated posterior approximations have been shown to be functionally competitive (e.g. Sheikh et al., 2014; Hughes & Sudderth, 2016), and they are also able to efficiently train large-scale models with hundreds or thousands of latents (e.g. Sheikh & Lücke, 2016; Forster & Lücke, 2018). In all these previous applications, optimization of truncated variational distributions relied on expectation maximization and closed-form or pseudo-closed form M-steps for shallow models. In the context of discrete latent VAEs, the important question arising is if or how the optimization of truncated variational distributions for the encoding model can be performed jointly with gradient-based optimization of the decoding model (which uses a DNN).

**Optimization of the Encoding Model.** Encoder optimization is usually based on a reformulation of the variational bound (2) given by:

$$\mathcal{F}(\Phi, \Theta) = \sum_n \mathbb{E}_{q_\Phi^{(n)}} \big[ \log \big( p_\Theta(\vec{x}^{(n)} \,|\, \vec{z}) \big) \big] - \sum_n D_{\mathrm{KL}} \big[ q_\Phi^{(n)}(\vec{z}); p_\Theta(\vec{z}) \big]. \tag{3}$$

For discrete latent VAEs, the variational distributions in (3) are commonly replaced by an amortized encoding model $q_\Phi(\vec{z})$ with a DNN-based parameterization. When expectations w.r.t. $q_\Phi(\vec{z})$ are approximated (as usual) via sampling, the encoder optimization requires gradient estimation methods for discrete random variables (cf. Related Work and App. B.6). In contrast, our proposed approach uses truncated posteriors as variational distributions and hence deviates substantially from the aforementioned standard approach. Given a data point $\vec{x}^{(n)}$, a truncated posterior is simply the posterior itself truncated to a subset $\Phi^{(n)}$ of the latent space, i.e., for $\vec{z} \in \Phi^{(n)}$ applies:

$$q_\Phi^{(n)}(\vec{z}) := \frac{p_\Theta(\vec{z} \,|\, \vec{x}^{(n)})}{\sum\limits_{\vec{z}' \in \Phi^{(n)}} p_\Theta(\vec{z}' \,|\, \vec{x}^{(n)})} = \frac{p_\Theta(\vec{x}^{(n)} \,|\, \vec{z}) \, p_\Theta(\vec{z})}{\sum\limits_{\vec{z}' \in \Phi^{(n)}} p_\Theta(\vec{x}^{(n)} \,|\, \vec{z}') \, p_\Theta(\vec{z}')} \tag{4}$$

while $q_\Phi^{(n)}(\vec{z}) = 0$ for $\vec{z} \notin \Phi^{(n)}$. The subsets $\Phi = \{\Phi^{(n)}\}_{n=1}^N$ are the variational parameters. Centrally for this work, truncated posteriors allow for a specific alternative reformulation of the bound. The reformulation recombines the entropy term of the original form (2) with the first expectation value into a single term, and is given by (see Lücke & Forster, 2019; Lücke, 2019, for details):

$$\mathcal{F}(\Phi, \Theta) = \sum_n \log \big( \sum_{\vec{z} \in \Phi^{(n)}} p_\Theta(\vec{x}^{(n)} \,|\, \vec{z}) \, p_\Theta(\vec{z}) \big). \tag{5}$$

Thanks to the simplified form of the bound, the variational parameters $\Phi^{(n)}$ of the encoding model can now be sought using direct discrete optimization procedures. More concretely, because of the specific form (5), pairwise comparisons of joint probabilities are sufficient to maximize the lower bound: if we update the set $\Phi^{(n)}$ for a given $\vec{x}^{(n)}$ by replacing a state $\vec{z}^{\mathrm{old}} \in \Phi^{(n)}$ with a state $\vec{z}^{\mathrm{new}} \notin \Phi^{(n)}$, then $\mathcal{F}(\Phi, \Theta)$ increases if and only if:

$$\log \big( p_\Theta(\vec{x}^{(n)}, \vec{z}^{\mathrm{new}}) \big) > \log \big( p_\Theta(\vec{x}^{(n)}, \vec{z}^{\mathrm{old}}) \big). \tag{6}$$

To obtain intuition for the pairwise comparison, consider its form when inserting the binary VAE (1) into the left- and right-hand sides. Eliding terms that do not depend on $\vec{z}$ we obtain:

$$\widetilde{\log p_\Theta}(\vec{x}, \vec{z}) = -\|\vec{x} - \vec{\mu}(\vec{z}, W)\|^2 - 2\sigma^2 \sum_h \tilde{\pi}_h z_h \tag{7}$$

where $\tilde{\pi}_h = \log \big( (1 - \pi_h)/\pi_h \big)$. The expression assumes an even more familiar form if we restrict ourselves for a moment to sparse priors with $\pi_h = \pi < \frac{1}{2}$, i.e., $\tilde{\pi}_h = \tilde{\pi} > 0$. Criterion (6) then becomes:

$$\|\vec{x}^{(n)} - \vec{\mu}(\vec{z}^{\mathrm{new}}, W)\|^2 + 2\sigma^2\tilde{\pi} \,|\vec{z}^{\mathrm{new}}| < \|\vec{x}^{(n)} - \vec{\mu}(\vec{z}^{\mathrm{old}}, W)\|^2 + 2\sigma^2\tilde{\pi} \,|\vec{z}^{\mathrm{old}}|, \tag{8}$$

where $|\vec{z}| = \sum_{h=1}^H z_h$. Such functions are routinely encountered in sparse coding or compressive sensing (Eldar & Kutyniok, 2012): for each set $\Phi^{(n)}$ we seek those states $\vec{z}$ that are reconstructing $\vec{x}^{(n)}$ well while being sparse ($\vec{z}$ with few non-zero bits). For VAEs, $\vec{\mu}(\vec{z}, W)$ is a DNN and as such much more flexible in matching the distribution of observables $\vec{x}$ than can be expected from linear mappings. Furthermore, criteria like (8) usually emerge for maximum a-posteriori (MAP) training

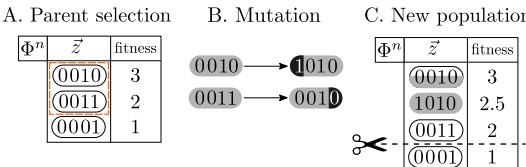

Figure 1: The optimization process of the variational parameters $\Phi^{(n)}$ using evolutionary search. **A.** Some states are selected as parents. **B.** Each child undergoes mutation. **C.** Children are merged with the original population and the least fit are discarded.

in sparse coding (Olshausen & Field, 1996). In contrast to MAP, however, we here seek a *population* of states $\vec{z}$ in $\Phi^{(n)}$ for each data point. It is a consequence of the reformulated lower bound (5) that it remains optimal to evaluate joint probabilities (as for MAP) although the constructed population of states $\Phi^{(n)}$ can capture (unlike MAP training) a rich posterior structure.

*Evolutionary Search.* But how can new states $\vec{z}^{\text{new}}$ that optimize $\Phi^{(n)}$ be found efficiently in high-dimensional latent spaces? Random search and search by sampling has recently been explored for elementary generative models (Lücke et al., 2018). Here we will follow another recent suggestion (Guiraud et al., 2018) and make use of a search based on evolutionary algorithms (EAs). In this setting we interpret sets $\Phi^{(n)}$ as populations of binary genomes $\vec{z}$ and base the fitness function on Eqn. (7). Concretely, using $\Phi^{(n)}$ as initial parent pool, we apply the following genetic operators in sequence to suggest new states to update the $\Phi^{(n)}$ based on Eq. 6 (see Fig. 1 for an illustration and App. A.1 for further details): firstly, *parent selection* stochastically picks states from the parent pool. Each of these states undergoes *mutation*: one or more bits are flipped to further increase offspring diversity. Crossover could also be employed to increase offspring diversity. We repeat the procedure using the *children* generated this way as the parent pool, giving birth to multiple *generations* of candidate states. Finally, we update $\Phi^{(n)}$ by substituting individuals with low fitness with candidates with higher fitness. The whole procedure can be seen as an evolutionary algorithm with perfect memory or very strong elitism (individuals with higher fitness never drop out of the gene pool). Note that the improvement of the variational lower bound depends on generating as many as possible *different* children with high fitness over the course of training.

We point out that the EAs optimize each $\Phi^{(n)}$ independently, which allows for distributed execution s.t. the technique can be efficiently applied to large datasets in conjunction with stochastic or batch gradient descent on the model parameters $\Theta$. The approach is, at the same time, memory intensive, i.e. all sets $\Phi^{(n)}$ need to be kept in memory (see App. A for details).

**Optimization of the Decoding Model.** Using the previously described encoding model $q_\Phi^{(n)}(\vec{z})$, we can compute the gradient of (2) w.r.t. the decoder weights $W$ which results in (see App. A for more details):

$$\vec{\nabla}_W \mathcal{F}(\Phi, \Theta) = -\frac{1}{2\sigma^2} \sum_n \sum_{\vec{z} \in \Phi^{(n)}} q_\Phi^{(n)}(\vec{z}) \; \vec{\nabla}_W \|\vec{x}^{(n)} - \vec{\mu}(\vec{z}, W)\|^2 \tag{9}$$

The right-hand-side has salient similarities to standard gradient ascent for VAE decoders. Especially the familiar gradient of the mean squared error (MSE) shows that, e.g., standard automatic differentiation tools can be applied. However, the decisive difference are the weighting factors $q_\Phi^{(n)}(\vec{z})$. Considering (4), in order to compute the weighting factors we require all $\vec{z} \in \Phi^{(n)}$ to be passed through the decoder DNN. As all states of $\Phi^{(n)}$ anyway have to be passed through the decoder for the MSE term of (9), the overall computational complexity is not higher than an estimation of the gradient with samples instead of states in $\Phi^{(n)}$ (but we use many states per $\Phi^{(n)}$, compare Tab. 1).

To complete the decoder optimization, update equations for variance $\sigma^2$ and prior parameters $\vec{\pi}$ can be computed in closed-form (compare, e.g., Shelton et al., 2011) and are given by

$$\sigma^{2,\text{new}} = \frac{1}{DN} \sum_n \sum_{\vec{z} \in \Phi^{(n)}} q_\Phi^{(n)}(\vec{z}) \, \|\vec{x}^{(n)} - \vec{\mu}(\vec{z}, W)\|^2 \;\; \text{and} \;\; \vec{\pi}^{\text{new}} = \frac{1}{N} \sum_n \sum_{\vec{z} \in \Phi^{(n)}} q_\Phi^{(n)}(\vec{z}) \, \vec{z}. \tag{10}$$

The full training procedure for binary VAEs is summarized in Algorithm 1. We refer to the binary VAE trained with this procedure as *Truncated Variational Autoencoder* (TVAE) because of the applied truncated posteriors.

**Algorithm 1** Training Truncated Variational Autoencoders

Initialize model parameters $\Theta = \{W, \vec{\pi}, \sigma^2\}$

Initialize each $\Phi^{(n)}$ with $S$ distinct latent states

**repeat**

    **for all** batches in dataset **do**

        **for** sample $n$ in batch **do**

            $\Phi^{new} = \Phi^{(n)}$

            **for all** generations **do**

                $\Phi^{new} = $ mutation (selection ($\Phi^{new}$))

                $\Phi^{(n)} = \Phi^{(n)} \cup \Phi^{new}$

            **end for**

            Truncate $\Phi^{(n)}$ to $S$ fittest elements based on (7)

        **end for**

        Use Adam to update $W$ using (9)

    **end for**

    Use (10) to update $\vec{\pi}, \sigma^2$

**until** parameters $\Theta$ have sufficiently converged

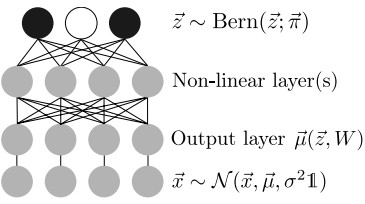

$\vec{z} \sim \mathrm{Bern}(\vec{z}; \vec{\pi})$

Non-linear layer(s)

Output layer $\vec{\mu}(\vec{z}, W)$

$\vec{x} \sim \mathcal{N}(\vec{x}, \vec{\mu}, \sigma^2 \mathbb{1})$

Figure 2: Graphical Representation of the Model Architecture Used in Numerical Experiments.

## 3 NUMERICAL EXPERIMENTS

TVAE can flexibly learn prior parameters $\pi_1, \ldots, \pi_H$, and if low values for the $\pi_h$ are obtained (which will be the case) the code is sparse. The prototypical application domain to study sparse codes is image patch data (Olshausen & Field, 1996; Goodfellow et al., 2012). We consequently use such data to investigate sparsity, scalability and efficiency on benchmarks. In App. B.7, we furthermore report results for a benchmark with audio data. For all numerical experiments, we employ fully connected DNNs $\vec{\mu}(\vec{z}; W)$ for the decoder as illustrated in Fig. 2; the exact network architectures and activations used are listed in Tab. 1. The DNN parameters are optimized based on (9) using mini-batches and the Adam optimizer (details in App. B.1)

**Sparsity and scalability.** After first verifying that the procedure can recover generating parameters using ground-truth data (see App. B.2), we used $N = 100,000$ whitened image patches of $16 \times 16$ pixels (van Hateren & van der Schaaf, 1998). We then applied TVAE with different settings. First, we used a model with $H = 300$ and a linear decoder; and, second, a TVAE with $H = 300$ and a deep decoder with architecture (300-300-256), see App. B.3 for details. Regarding sparsity, we observed for linear and non-linear TVAEs a sparse encoding. For the linear TVAE we observed on average 20.3 latents active across data points (i.e., $\sum_h \pi_h / H = 20.3 / 300$), and for the non-linear TVAE we observed on average 28.5 latents to be active ($\sum_h \pi_h / H = 28.5 / 300$). We observed sparse codes also when we changed parameter initialization, used other DNNs and when changing latent dimensions. Furthermore, we observed efficient scalability to large latent spaces (we went up to $H = 1000$) as long as decoder DNNs were of small to intermediate size. Compared to linear decoders, the main additional computational cost is given by passing the latent states of $\Phi^{(n)}$ through the decoder DNN. instead of just through a linear mapping. The sets of states (i.e., the bitvectors in $\Phi^{(n)}$) could be kept small, at size $S = |\Phi^{(n)}| = 64$, such that $N \times (|\Phi^{(n)}| + |\Phi^{(n)}_{\mathrm{new}}|)$ states had to be evaluated for each epoch. This compares to $N \times M$ states that would be used for standard VAE training (given $M$ samples are drawn per data point). Differently to standard VAE training the sets $\Phi^{(n)}$ have to be remembered across iterations. For very large datasets, the additional $\mathcal{O}(N \times |\Phi^{(n)}| \times H)$ memory demand can be distributed over compute nodes, however.

**Denoising and inpainting.** The experiments above show a notable difference in scalability characteristics compared to other (including discrete) VAEs. Using a non-amortized approaches means that memory and computational load more strongly increase with data points than for amortized training. Above, we processed $100,000$ data points which is still feasible for DNNs of intermediate size. Large decoder DNNs increase the computational load significantly, however, as $N \times (|\Phi^{(n)}| + |\Phi^{(n)}_{\mathrm{new}}|)$ latent states have to be passed through the decoder. Furthermore, larger DNNs require more data points to not overfit which further increases computational load of our $N$-dependent method. Tasks such as disentanglement of features using high-dimensional input data, large DNNs, and small latent spaces (see introduction) are, therefore, not a regime where the here studied approach can be applied efficiently. However, for tasks with relatively few data for which an as effective as possible optimization is required, advantages of direct optimization can be expected.

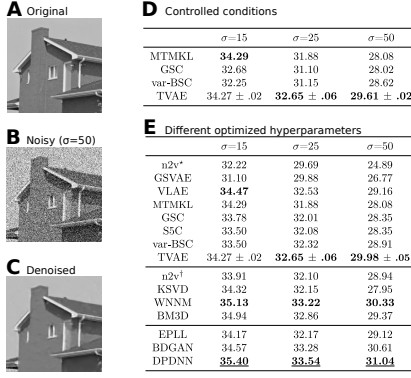

**A** Original

**B** Noisy ($\sigma$=50)

**C** Denoised

**D** Controlled conditions

| | $\sigma$=15 | $\sigma$=25 | $\sigma$=50 |
|---|---|---|---|
| MTMKL | 34.29 | 31.88 | 28.08 |
| GSC | 32.68 | 31.10 | 28.02 |
| var-BSC | 32.25 | 31.15 | 28.62 |
| TVAE | 34.27 ± .02 | 32.65 ± .06 | 29.61 ± .02 |

**E** Different optimized hyperparameters

| | $\sigma$=15 | $\sigma$=25 | $\sigma$=50 |
|---|---|---|---|
| n2v* | 32.22 | 29.69 | 24.89 |
| GSVAE | 31.10 | 29.88 | 26.77 |
| VLAE | 34.47 | 32.53 | 29.16 |
| MTMKL | 34.29 | 31.88 | 28.08 |
| GSC | 33.78 | 32.01 | 28.35 |
| S5C | 33.50 | 32.08 | 28.35 |
| var-BSC | 33.50 | 32.32 | 28.91 |
| TVAE | 34.27 ± .02 | 32.65 ± .06 | 29.98 ± .05 |
| n2v† | 33.91 | 32.10 | 28.94 |
| KSVD | 34.32 | 32.15 | 27.95 |
| WNNM | 35.13 | 33.22 | 30.33 |
| BM3D | 34.94 | 32.86 | 29.37 |
| EPLL | 34.17 | 32.17 | 29.12 |
| BDGAN | 34.57 | 33.28 | 30.61 |
| DPDNN | 35.40 | 33.54 | 31.04 |

Figure 3: Denoising results for House. **D** compares PSNRs (in dB) obtained with different 'zero-shot' models using a fixed patch size and number of latents (see text for details). **E** lists PSNRs for different algorithms with different optimized hyper-parameters. The top category only requires the noisy image. The middle requires additional information such as noise level (KSVD, WNNM, BM3D) or additional noisy images with matched noise level (n2v†). The bottom three algorithms use large clean datasets. **C** depicts the denoised image obtained with TVAE for $\sigma = 50$ in the best run (PSNR=30.03 dB).

For image patches, sparse codes are often preferred to dense encoding. A canonical benchmark to evaluate image patch models is 'zero-shot' denoising, which recently became popular also because conventional DNN denoising is not offering good solutions. 'Zero-shot' means that only the noisy image is used for training. If no clean data is available, variations of feed-forward DNNs have been suggested whose training objectives have been altered to enable training on single (or few) noisy images (e.g. Lehtinen et al., 2018; Krull et al., 2019b). However, deep *generative* models are (we would argue) more natural candidates to train on noisy data as their learning objective can be used directly. If few data is available for 'zero-shot' learning, large and very deep DNNs cannot be used. Shocher et al. (2018), for instance, also argue that smaller DNNs are sufficient for the 'zero-shot' setting. The task is consequently natural and well suited to evaluate direct optimization, and it has the very significant additional benefit of allowing for a comparison to a large range of other approaches that have recently been suggested. Most notably, we can compare to other VAEs (Jang et al., 2016; Park et al., 2019a), to sparse coding, to large feed-forward DNNs (Zhu et al., 2019; Dong et al., 2019), and to DNNs dedicated to learning from noisy data (Lehtinen et al., 2018; Krull et al., 2019b). Furthermore, we evaluate our approach on image inpainting. Like denoising, inpainting allows for comparison to other reported results on standard benchmarks. Additionally, inpainting highlights another advantage of the direct optimization approach we study: as no encoder DNNs are used, missing data can naturally and directly be treated probabilistically (see below).

The one denoising benchmark that offers the broadest possible comparison to other methods is the 'house' image (Fig. 3 A). Standard benchmark settings for 'house' make use of additive Gaussian white noise with standard deviations $\sigma \in \{15, 25, 50\}$. First, consider the comparison in Fig. 3 D where all models used the same patch size of $D = 8 \times 8$ pixels and $H = 64$ latent variables (App. B.4 for details). Fig. 3 D lists the different approaches in terms of the standard measure of peak signal-to-noise ratio (PSNR). Values for MTMKL (Titsias & Lázaro-Gredilla, 2011), GSC (Sheikh et al., 2014) and S5C (Sheikh & Lücke, 2016) were taken from the respective original publications (which all established new state-of-the-art results when first published). As can be observed, TVAE significantly improves performance for high noise levels. The approach is able to learn the best data representation for denoising and represents the state-of-the-art in this controlled setting (i.e., fixed $D$ and $H$). The decoder DNN of TVAE provides the decisive performance advantage: TVAE significantly improves performance compared to linear Binary Sparse Coding (var-BSC, Henniges et al., 2010; Shelton et al., 2011), confirming that the high lower bounds of TVAE on natural images translate into improved performance on a concrete benchmark. For $\sigma = 25$ and $\sigma = 50$, TVAE also significantly improves on MTMKL, GSC, and S5C. These three approaches are based on a spike-and-slab sparse coding model (also compare Goodfellow et al., 2012). Despite the less flexible Bernoulli prior, the decoder DNN of TVAE provides the highest PSNR values for high noise levels.

To extend the comparison, we next considered the denoising task without controlling for equal conditions. Concretely, we allowed for any approach that performs denoising on the benchmark including approaches that are trained on large image datasets and/or use different patch sizes (including multi-scale and whole image processing). Note that different approaches may employ very different sets of hyper-parameters that can be optimized for denoising performance: for sparse coding approaches, hyper-parameters include patch and dictionary sizes; for DNN approaches they include all network and training scheme hyper-parameters. By allowing for comparison in this less controlled setting, we can include a number of recent approaches including large DNNs trained

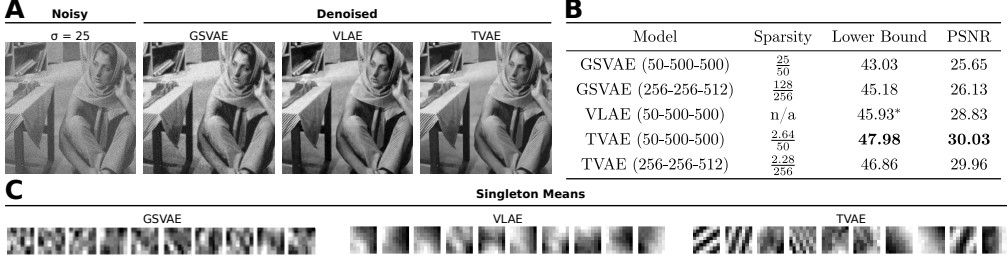

| Model | Sparsity | Lower Bound | PSNR |
|---|---|---|---|
| GSVAE (50-500-500) | $\frac{25}{50}$ | 43.03 | 25.65 |
| GSVAE (256-256-512) | $\frac{128}{256}$ | 45.18 | 26.13 |
| VLAE (50-500-500) | n/a | 45.93* | 28.83 |
| TVAE (50-500-500) | $\frac{2.64}{50}$ | **47.98** | **30.03** |
| TVAE (256-256-512) | $\frac{2.28}{256}$ | 46.86 | 29.96 |

Figure 4: Comparison of data representations learned by denoising image patches (cf. App. B.6).

on clean data and training schemes specifically targeted to noisy training data. Fig. 3 E shows the denoising performance for the three noise levels we investigated, with results for other algorithms taken from their corresponding original publications unless specified otherwise. For WNNM and EPLL we cite values from Zhang et al. (2017). The results reported for noise2void (n2v, Krull et al., 2019b), Gumbel-softmax VAE (GSVAE; Jang et al., 2016), VLAE (Park et al., 2019a), we produced ourselves by applying publicly available source code (see App. B.4).

Note that the best performing approaches in Fig. 3 E were trained on noiseless data: EPLL (Zoran & Weiss, 2011), BDGAN (Zhu et al., 2019) and DPDNN (Dong et al., 2019) all make use of clean training data (typically hundreds of thousands of data points or more). For denoising, EPLL also requires the ground-truth noise level of the test image. Ground-truth noise level information is also required by KSVD (Elad & Aharon, 2006) and WNNM (Gu et al., 2014). As noisy data is very frequently occurring, lifting the requirement of clean data has been of considerable recent interest with, e.g., approaches like noise2noise (n2n Lehtinen et al., 2018) and noise2void having received considerable attention. The n2n approach can achieve denoising performance on noisy training data which is almost as high as the performance of a given DNN when trained on clean data. It would thus outperform all approaches in Fig. 3 E except for the bottom three. However, n2n requires different noise realizations of the very same underlying image. The noise2void approach aims to remove this unrealistic, artificial assumption. Considering Fig. 3 E, PSNR values of TVAE were consistently higher than those of n2v even if n2v was trained on external data with matched-noise level (n2v[†] in Fig. 3 E). Performance of TVAE is 0.2dB lower than BM3D for $\sigma = 25$ and 0.6dB higher for $\sigma = 50$, which makes it, for larger noise levels, the state-of-the-art on this benchmark in the 'zero-shot' setting (i.e., the setting n2n and n2v aim to address).

Besides measuring PSNRs, we investigated the generative representation that TVAE used for denoising. For TVAE, training and data reconstruction is performed based on patches taken from the noisy image, and reconstructed image patches are finally combined to generate the denoised image (see App. B.4 for details). Based on such image patches, TVAE learns a sparse encoding which, in terms of lower bounds, shows to be very competitive compared to other generative models (compare Fig. 4). For the House denoising benchmark (Fig. 3 E), the sparsity, i.e., the average number of active latents $\sum_h \pi_h / H$ was 5.98 / 64 ($\sigma = 15$), 4.47 / 64 ($\sigma = 25$), 1.65 / 512 ($\sigma = 50$).

Finally, TVAE can be applied to 'zero-shot' inpainting tasks using a procedure similar to the one employed for denoising. For TVAE, the treatment of missing data is directly available. Missing pixel values are simply treated as such using the probabilistic description of the model. Concretely, when evaluating log-joint probabilities of a data-point during training, missing values are treated as unknown observables (App. B.5). In contrast, amortized approaches will have to specify how the deterministic encoder DNNs should treat missing values. Fig. 5 compares PSNRs obtained with TVAE to other methods: the left table shows results for 'House', where we separate BPFA and TVAE, which are permutation-invariant approaches, from DIP and Papyan et al. (2017), which are not permutation-invariant. Fig. 5 (right) shows results for 'Castle', where we separate 'zero-shot' approaches (top section) from non-'zero-shot' approaches (bottom section; PLE refers to the approach from Yu et al. (2012), IRCNN refers to Chaudhury & Roy (2017)). Fig. 5 (right) shows the result of 'zero-shot' inpainting on 'Castle' with 50% missing pixels (more details in App. B.5). As can be observed, TVAE is state-of-the-art for 'House' and for the larger 'Castle' with 50% missing pixels. But if non-permutation invariant DNNs with specific, large U-nets are used (DIP for 'House'), or if models are trained on clean data, then higher PSNR values are achieved.

| | House 50% |
|---|---|
| MTMKL | n/a |
| BPFA | 38.02 |
| TVAE | 38.56 |
| Papyan et al. | 34.58 |
| DIP | **39.16** |

| | Castle 50% | Castle 80% |
|---|---|---|
| MTMKL | n/a | 28.94 |
| BPFA | 36.45 | **29.12** |
| TVAE | **37.33** | 28.93 |
| PLE | **38.34** | **30.07** |
| IRCNN | n/a | 28.74 |

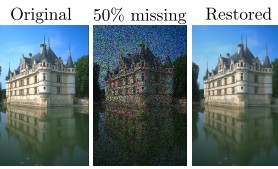

Original  50% missing  Restored

Figure 5: Inpainting results for House (50% missing pixels) and Castle (50% and 80% missing). Highest PSNR in the 'zero-shot' setting (top rows) marked bold; overall highest PSNRs are bold and underlined. PLE requires the noise level as additional information; IRCNN requires external clean training images; Papyan et al. and DIP are not permutation invariant. The rightmost image depicts TVAE's inpainting result for Castle (50% missing; sparsity $\sum_h \pi_h / H$ was 10.56 / 512).

## 4 DISCUSSION

We investigated VAEs with binary priors able to learn sparse latent codes. We were in general interested in how such codes can efficiently be learned, and what advantages (and disadvantages) can be observed. Compared to all previous VAEs with discrete latents, the approach we followed differs the most substantially from conventional VAE learning. While all other VAEs maintain amortization and reparameterization as key elements, the TVAE approach instead uses a direct discrete optimization of binary latent vectors. A conceptual advantage of the approach is its concise formulation (compare Fig. 7) with fewer algorithmic elements, fewer hyperparameters and fewer model parameters (e.g., no parameters of encoder DNNs). Functional advantages of the approach are its avoidance of an amortization gap, its ability to learn sparse codes, and its generality (it does not use a specific posterior model, and can be applied to other noise models, for instance). However, the disadvantage of a non-amortized approach is a lower computational efficiency: we optimize variational parameters for each data point which is more costly. Conventional amortized approaches (for discrete or continuous VAEs) are consequently preferable if one seeks to optimize large, intricate DNNs on large data sets. There are, however, alternatives such as transformers (which can use >150M parameters) or diffusion nets, which both are considered to perform more strongly than VAEs for large-scale settings and density modeling (Child et al., 2019; Kingma et al., 2021, for recent comparisons).

When no sufficiently large datasets are available, models incorporating large DNNs can not be optimized. In such cases, we observe sparse codes obtained by direct discrete optimization to be advantageous. For image patch modeling, TVAE with intermediately large decoder DNNs outperformed Gumbel-softmax VAEs (GSVAE) as well as, e.g., a recent continuous VAE baseline (VLAE; Park et al., 2019a, Figs. 3,4). The competitive performance is presumably due to the approach not being subject to an amortization gap, due to it avoiding factored variational distributions, and more generally due to the emerging sparse codes being well suited to model image patches. In comparison, the additional methods to treat discrete latents in GSVAE seem to result in dense codes with significantly lower performance than TVAE. Performance of VLAE is much more competitive. While VLAE uses a vanilla non-sparse (i.e. Gaussian) prior, its continuous encoding is presumably a reason why its dense codes result in higher ELBO values than GSVAE (but in lower values than TVAE). Dense codes are notably not necessarily disadvantages for image data. For datasets with many images of single objects like CIFAR, the dense code of GSVAE and also of VLAE is similar or better in terms of ELBO values than TVAE (Fig. 15). The suitability of sparse vs. dense codes consequently seems to highly depend on the data, and here we confirm the suitability of sparse codes for image patches. Based on the high ELBO values of TVAE for patches (Figs. 4,12), we observed TVAE to result in state-of-the-art performance on the recently popular task of 'zero-shot' denoising (also see Fig. 16 for benchmarking results with audio data). Likewise, the approach is competitive for 'zero-shot' inpainting. Inpainting highlights another advantage of direct optimization: in contrast to other (continuous or discrete) VAEs, it is not required to additionally specify how missing data shall be treated by an encoder DNN (see App. B.5).

Our conclusion is consequently that direct discrete optimization can, depending on the task, serve as an alternative for training discrete VAEs. In a sense, the approach can be considered more brute-force than conventional amortized training: direct optimization is slower but at scales at which it can be applied more effective. To our knowledge, the approach is also the first training method for discrete VAEs not using gradient optimization of encoder models, and in general the first which makes discrete VAEs competitive, e.g., for 'zero-shot' denoising and inpainting.

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

# Supplementary Material

## A    DETAILS OF ENCODER AND DECODER OPTIMIZATION

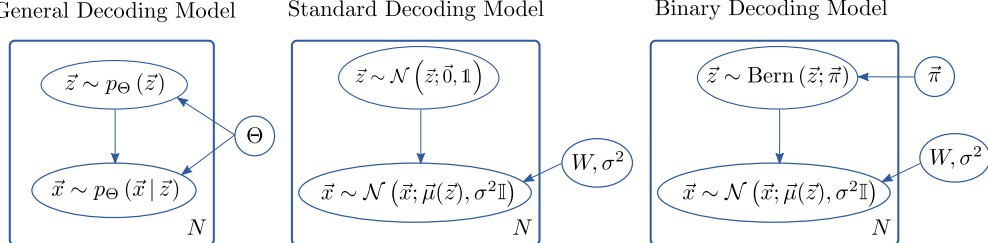

Figure 6: Generic and general VAE decoding model (left), VAE model with standard continuous latents (center), and the VAE model with binary latents (right) of Eqn. (1) (right).

See Fig. 6 for a graphical comparison between the decoding models of a vanilla VAE and the binary VAE considered here (1). Fig. 7 graphically illustrates different steps to optimize standard VAEs, and additional steps suggested by different contributions in order to optimize discrete VAEs.

For the optimization of the binary VAE (1), consider the original form of the lower bound, Eqn. (2). When taking derivatives of $\mathcal{F}(\Phi, \Theta)$ w.r.t. $\Theta$ we can ignore the entropy term[1]. For the binary VAE model of Eqn. (1) the gradient of the lower bound w.r.t. $W$ is then given by:

$$
\begin{aligned}
\vec{\nabla}_W \mathcal{F}(\Phi, \Theta) &= \sum_n \vec{\nabla}_W \mathbb{E}_{q_\Phi^{(n)}}\big[\log\big(p_\Theta(\vec{x}^{(n)} \,|\, \vec{z})\, p_\Theta(\vec{z})\big)\big] \\
&= \sum_n \vec{\nabla}_W \mathbb{E}_{q_\Phi^{(n)}}\big[\log\big(p_\Theta(\vec{x}^{(n)} \,|\, \vec{z})\big)\big] \\
&= \sum_n \vec{\nabla}_W \mathbb{E}_{q_\Phi^{(n)}}\big[\log\big(\mathcal{N}(\vec{x}^{(n)}; \vec{\mu}(\vec{z}, W), \sigma^2 \mathbb{I})\big)\big] \\
&= -\frac{1}{2\sigma^2} \sum_n \vec{\nabla}_W \sum_{\vec{z} \in \Phi^{(n)}} q_\Phi^{(n)}(\vec{z})\, \|\vec{x}^{(n)} - \vec{\mu}(\vec{z}, W)\|^2 \\
&= -\frac{1}{2\sigma^2} \sum_n \sum_{\vec{z} \in \Phi^{(n)}} q_\Phi^{(n)}(\vec{z})\, \vec{\nabla}_W \|\vec{x}^{(n)} - \vec{\mu}(\vec{z}, W)\|_2^2; \quad (11)
\end{aligned}
$$

where, by using (4) and (1), the weighting factors $q_\Phi^{(n)}(\vec{z})$ are given by:

$$
\begin{aligned}
q_\Phi^{(n)}(\vec{z}) &= \frac{p_\Theta(\vec{x}^{(n)} \,|\, \vec{z})\, p_\Theta(\vec{z})}{\sum_{\vec{z}' \in \Phi^{(n)}} p_\Theta(\vec{x}^{(n)} \,|\, \vec{z}')\, p_\Theta(\vec{z}')} \\
&= \frac{\exp\big(-\frac{1}{2\sigma^2}\, \|\vec{x}^{(n)} - \vec{\mu}(\vec{z}, W)\|^2 - \tilde{\vec{\pi}}^T \vec{z}\big)}{\sum_{\vec{z}' \in \Phi^{(n)}} \exp\big(-\frac{1}{2\sigma^2}\, \|\vec{x}^{(n)} - \vec{\mu}(\vec{z}', W)\|^2 - \tilde{\vec{\pi}}^T \vec{z}'\big)}
\end{aligned} \quad (12)
$$

for all $\vec{z} \in \Phi^{(n)}$, where $\tilde{\pi}_h = \log\big(\frac{1 - \pi_h}{\pi_h}\big)$. Note that the $q_\Phi^{(n)}(\vec{z})$ are evaluated at the current values of the parameters $\Theta$, they are therefore treated as constant, e.g., for the gradient w.r.t. $W$.

It may be interesting to compare the gradient estimate (11) to the gradient estimate of conventional VAE training. For this consider a standard encoder given by an amortized variational distribution

---

[1]For our choice of variational distributions, it is not trivial that the entropy term actually can be ignored because the encoding model $q_\Phi(\vec{z}; \vec{x})$ in (4) is defined in terms of the decoding model and its parameters. For truncated distributions, however, it can be shown that the entropy term can still be ignored (Lücke, 2019).

which we shall denote by $\tilde{q}_\Phi^{(n)}(\vec{z})$. The distribution $\tilde{q}_\Phi^{(n)}(\vec{z})$ could be a Gaussian whose mean and variance are set by passing data point $\vec{x}^{(n)}$ through encoder DNNs. For discrete VAEs, $\tilde{q}_\Phi^{(n)}(\vec{z})$ can be thought of as an analog discrete distribution. If we now take gradients of (3) w.r.t. $W$ and estimate using samples from $\tilde{q}_\Phi^{(n)}(\vec{z})$, we obtain the familiar form:

$$\vec{\nabla}_W \mathcal{F}(\Phi, \Theta) = \sum_n \vec{\nabla}_W \mathbb{E}_{\tilde{q}_\Phi^{(n)}} \left[ \log \left( p_\Theta(\vec{x}^{(n)} \mid \vec{z}) \, p_\Theta(\vec{z}) \right) \right]$$

$$= \sum_n \vec{\nabla}_W \mathbb{E}_{\tilde{q}_\Phi^{(n)}} \left[ \log \left( \mathcal{N}(\vec{x}^{(n)}; \vec{\mu}(\vec{z}, W), \sigma^2 \mathbb{I}) \right) \right]$$

$$\approx -\frac{1}{2\sigma^2} \sum_n \frac{1}{M} \sum_{m=1}^M \vec{\nabla}_W \| \vec{x}^{(n)} - \vec{\mu}(\vec{z}^{(m)}, W) \|^2$$

where $\vec{z}^{(m)} \sim \tilde{q}_\Phi^{(n)}(\vec{z})$. We can slightly rewrite this expression to obtain:

$$\vec{\nabla}_W \mathcal{F}(\Phi, \Theta) \approx -\frac{1}{2\sigma^2} \sum_n \sum_{\vec{z} \sim \tilde{q}_\Phi^{(n)}} \left( \frac{1}{M} \right) \vec{\nabla}_W \| \vec{x}^{(n)} - \vec{\mu}(\vec{z}, W) \|^2 , \tag{13}$$

If we now compare with the gradient using the truncated approximation $q_\Phi^{(n)}(\vec{z})$,

$$\vec{\nabla}_W \mathcal{F}(\Phi, \Theta) = -\frac{1}{2\sigma^2} \sum_n \sum_{\vec{z} \in \Phi^{(n)}} q_\Phi^{(n)}(\vec{z}) \, \vec{\nabla}_W \| \vec{x}^{(n)} - \vec{\mu}(\vec{z}, W) \|^2 , \tag{14}$$

one can discuss analogous roles played by the subsets $\Phi^{(n)}$ (the variational parameters of $q_\Phi^{(n)}(\vec{z})$) and by a standard encoder $\tilde{q}_\Phi^{(n)}$. The states in a subset $\Phi^{(n)}$ are used to estimate the gradient similar to the samples from a standard encoder $\tilde{q}_\Phi^{(n)}(\vec{z})$. The size of $\Phi^{(n)}$ can consequently be thought of as analog to the number of samples used in a conventional estimation of the gradient. Standard VAE training estimates the gradient by weighting all samples equally (with $(1/M)$) and the gradient direction is approximated using sufficiently many samples drawn from the current $\tilde{q}_\Phi^{(n)}(\vec{z})$. In contrast, truncated gradient estimation uses the states in $\Phi^{(n)}$, and the gradient is computed using a weighted summation with weights $q_\Phi^{(n)}(\vec{z})$. These weights are computed by passing the states $\vec{z}$ through the *decoder* network. The gradient is then, notably, not a stochastic estimation but exact: gradient ascent is guaranteed (for small steps) to always monotonically increase the variational lower bound.

**Computational Complexity.** To add to the discussion of computational complexity of TVAE compared to standard VAE training, consider again Eqns. 13 and 14. If as many samples $M$ are used, per data point, as there are states in each $\Phi^{(n)}$, then both sums have the same number of summands. The evaluation of the gradients of the mean square error (MSE) is consequently precisely the same for both approaches. The additional weighting factors $q_\Phi^{(n)}(\vec{z})$ have to be computed for TVAE. However, the weighting factors just represent a small overhead because the evaluation of the decoder DNN for the states in $\Phi^{(n)}$ is a computation that can be reused from the updates of $\Phi^{(n)}$.

The main computational differences are in the updates of $\Phi^{(n)}$ compared to the update of encoder DNNs for conventional VAEs. Once the parameters $\Theta = (W, \sigma^2, \vec{\pi})$ are updated using (14), new states for $\Phi^{(n)}$ have to be sought based on criterion (7). In practice and for each $n$, we generate $M'$ new states according to the applied evolutionary procedure. To select the best states we have to pass all these $M'$ new states through the decoder DNN to evaluate (7). Furthermore, we have to pass all $M$ states already in $\Phi^{(n)}$ through the DNN to re-evaluate (7) because the parameters $\Theta$ have changed. In summary, we do require $\mathcal{O}(N \times (M + M'))$ passes through the decoder DNN. Selecting the $M$ best states from the $(M + M')$ states does not add complexity as this can be done in $\mathcal{O}(M + M')$ for each $n$ (Blum et al., 1973). The EA does add to the computational load but parent selection and mutation only add a constant offset for each of the considered states.

For comparison with standard VAEs, if we use $M$ samples of an encoder $\tilde{q}_\Phi^{(n)}(\vec{z})$, we require $\mathcal{O}(M \times N)$ passes through the decoder DNN to update the parameters $\Theta$ according to (13). For the encoder update, one requires $N \times \tilde{M}$ passes through encoder and decoder DNN to estimate the gradient

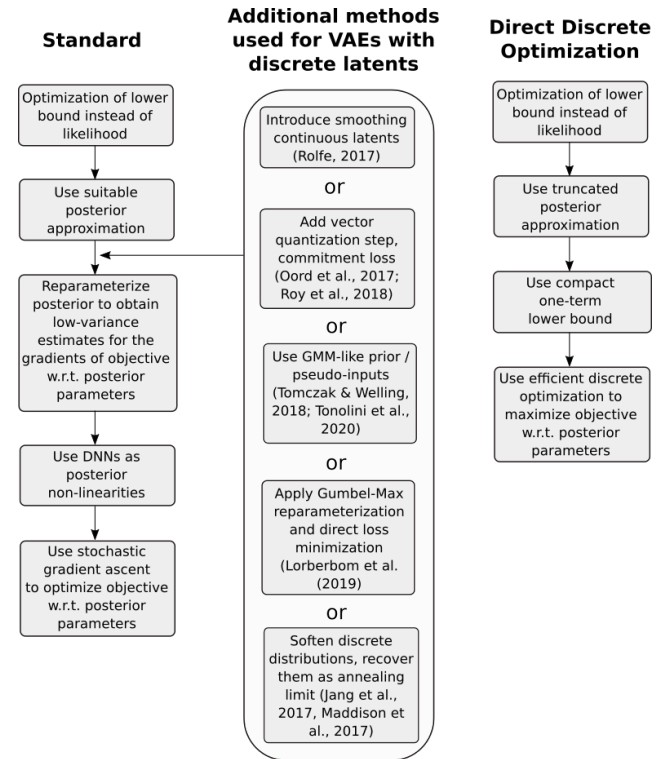

Figure 7: Typical collection of methods used to optimize the encoding model of VAEs. **Left:** Methods of the standard procedure to optimize VAEs. **Middle:** Examples of additional methods applied to maintain the standard VAE procedure also for VAEs with discrete latent variables. **Right:** Alternative direct discrete optimization of VAE encoding models.

w.r.t. the encoder weights (if we draw $\tilde{M}$ samples for each data point from a conventional encoder distribution $\tilde{q}_\Phi^{(n)}(\vec{z})$. The additional overhead to actually draw the samples is usually negligible.

Hence, the computational complexity of TVAE training is comparable if $M \approx M' \approx \tilde{M}$. However, conventional VAE training is amortized, i.e., the update of encoder weights uses information from all data points $n$. In contrast, TVAE training is not amortized, i.e., the $\Phi^{(n)}$ are updated per data point. The advantage of amortization is that in practice, weights of a conventional encoder can converge faster or (alternatively) less samples $\tilde{M}$ are required. Considering the observed runtimes, more efficient conventional VAE training can presumably in large parts attributed to faster convergence using amortization. Furthermore, the used number of samples $M$ for conventional VAE training is usually smaller than best working sizes of $\Phi^{(n)}$ (we used, e.g., $|\Phi^{(n)}| \in [20, 200]$ for denoising, see Tab. 1); and the required storage of $\Phi^{(n)}$ results in overhead computations. On the other hand, amortization also has disadvantages (e.g. Kim et al., 2018; Cremer et al., 2018). The competitive performance for denoising may consequently be attributed at least in part to TVAE not being subject to an amortization gap.

### A.1 EVOLUTIONARY ALGORITHMS

We base the fitness function on the joint $p_\Theta(\vec{x}, \vec{z})$ that follows from (1). Concretely we employ the reformulation (7), which allows for robust computation, and we use an offset (constant w.r.t. $\vec{z}$) to ensure that fitness values are strictly non-negative. We investigated different selection operators: (i) a uniform random selection procedure that is independent of the parents' fitness, and (ii) a procedure in which parents are selected with a probability proportional to their fitness. After selection, the

children states of a new generation are manipulated through mutation (and possibly crossover). The mutation operator flips one bit per child; the index of the bit to flip is uniformly randomized. To apply crossover, children states are paired in any possible way, and the tails of the states in each pair are swapped; the single crossover point is uniformly randomized.

In our large scale numerical experiments, we used fitness-proportional parent selection in combination with uniformly random bitflips, as we found this operator combination to efficiently and effectively suggest new states $\vec{z}^{\text{new}}$ for the optimization of the subsets $\Phi^{(n)}$ based on criterion (8) (cf. Fig. 11). We refer to this evolutionary operator design as *fitparents-randflips*; other operator designs we considered are termed accordingly (i.e., *randparents-cross-randflips*, for instance, indicates uniformly random parent selection, application of crossover and uniformly random bitflips; compare Fig. 11). Hyperparameters of the evolutionary algorithms are $N_p$ (denoting the number of parental states selected per generation; $N_p \leq |\Phi^{(n)}|$), $N_c$ (denoting the number of children evolved per parent), and $N_g$ (denoting the number of generations evolved). When crossover is employed, the total number of new states evolved per data point per epoch is given by $N_p(N_p - 1)N_g$, otherwise it is $N_p N_c N_g$. The concrete hyperparameters used in the numerical experiments are listed in Tab. 1.

## B  DETAILS ON THE NUMERICAL EXPERIMENTS

### B.1  HYPERPARAMETERS

For all numerical experiments, DNN training using (9) was performed with mini-batches, the Adam optimizer (Kingma & Ba, 2014) and decaying or cyclical learning rate scheduling (Smith, 2017). Xavier/Glorot initialization (Glorot & Bengio, 2010) was used for the DNN weights, while biases were always zero-initialized. Parameters $\vec{\pi}$ and $\sigma^2$ were updated via Eqn. (10). $\vec{\pi}$ was initialized to $\frac{1}{H}$. $\sigma^2$ was initialized to 0.01 with the exception of the Barbara, CIFAR-10 and Audio datasets (Figs. 4, 15 and 16) for which we initialized $\sigma^2$ with the data variance. The $\Phi^{(n)}$ were initialized by drawing $\vec{z}$ from a Bernoulli distribution with $p(z_h = 1) = \frac{1}{H}$. Hyperparameter optimization was conducted manually, and for the more complex datasets, it also made use of black box Bayesian optimization based on Gaussian Processes (Nogueira, 2019) and BOHB (Falkner et al., 2018) using the hpbandster framework (Biedenkapp & Hutter, 2018). Tab. 1 provides an overview of the hyperparameters used in each of the reported experiments.

### B.2  VERIFICATION EXPERIMENTS

We first evaluated TVAE training on artificial datasets with known ground-truth parameters and log-likelihood, in order to verify the correct functioning of the algorithm and to investigate possible local optima effects. The dataset consisted of 500 4x4 images generated by linear superposition of vertical and horizontal bars (compare, e.g. Földiák, 1990; Hoyer, 2003; Guiraud et al., 2018), with a small amount of Gaussian noise. The DNN's input and middle layers had 8 units each. The $\Phi^{(n)}$ variational sets consisted of 64 hidden states each. Fig. 8 shows the evolution of the run that achieved the highest ELBO value out of ten. All parameters were correctly recovered, and the ELBO value was consistent with actual ground-truth log-likelihood.

Such an elementary test, however, can also be solved by linear models. In order to demonstrate that TVAEs can solve non-linear problems, taking advantage of the neural network non-linearity embedded in the generative model, we introduced correlations between pairs of bars: the bars combinations shown in the first two data points from the left in Fig. 9 were discouraged from appearing together. Again we selected the run with highest peak ELBO value out of ten. The model correctly learns that certain combinations of bars are much more unlikely than others, and correctly estimates their likelihood.

Fig. 10 offers some more insight into the correlated bars test experiment described. The left section of the figure shows the generative parameters for the dataset used: $W_0$ is the 8x8 weight matrix of the top-to-middle layer: this makes it so that the activation of the first latent variable inhibits activation of the second, and activation of the last latent variable inhibits activation of the last. Concretely, this results in a dataset where these specific bars combinations are discouraged from appearing. The weights $W_1$, visualized as 8 4x4 matrices, generate the actual bars. $\sigma^2$ was set to 0.01 and the dataset contained an average of two superimposing bars per data point ($\pi_h = 2/8$ for each $h$). The middle

Table 1: Hyperparameters used in the numerical experiments. The architecture of the decoding DNN is listed as $H_0$-$H_1$-...-$D$ with $H_0$ and $D$ indicating the number of Bernoulli latents and Gaussian observables respectively, and $H_1$-... denoting the number of hidden layer units. By default, we used ReLU activations in the hidden layers and a linear output layer. For Barbara and CIFAR-10, we used LeakyReLU instead of ReLU; for CIFAR-10, we additionally used a Sigmoid in the output layer. Min and max l.r. denote lower and upper learning rate boundaries and are, together with Epochs/Cycle, hyperparameters of the cyclical learning rate scheduler (cf. Smith, 2017). $|\Phi^{(n)}|$ denotes the number of distinct latents per data point (referred to as $S$ in Alg. 1). $^{\dagger}$ and $^{\ddagger}$ refer to the parameters used for $\sigma \in \{15, 25\}$ and $\sigma = 50$ in Fig. 3E, respectively.

| | Decoding Model | | | | | Encoding Model | |
|---|---|---|---|---|---|---|---|
| | $H_0$-$H_1$-...-$D$ | Min l.r. | Max l.r. | Epochs/Cycle | Batch Size | $|\Phi^{(n)}|$ | EA |
| Bars (Fig. 8) | 8-8-16 | 0.0001 | 0.05 | 20 | 32 | 64 | fit-randflip ($N_p = 3$, $N_c = 2$, $N_g = 1$) |
| Bars (Figs. 9, 10) | 8-8-16 | 0.0001 | 0.1 | 50 | 32 | $2^{H_0}$ | *exact E-step* |
| Bars (Fig. 11) | 6-8-9 | 0.0001 | 0.05 | 20 | 32 | 32 | *random sampling* ($N_{\text{new}} = 20$, $p(z_h = 1) = \frac{1}{H_0}$) |
| Bars (Fig. 11) | 6-8-9 | 0.0001 | 0.05 | 20 | 32 | 32 | fit-randflip ($N_p = 5$, $N_c = 4$, $N_g = 1$) |
| Bars (Fig. 11) | 6-8-9 | 0.0001 | 0.05 | 20 | 32 | 32 | rand-randflip ($N_p = 5$, $N_c = 4$, $N_g = 1$) |
| Bars (Fig. 11) | 6-8-9 | 0.0001 | 0.05 | 20 | 32 | 32 | fit-cross-randflip ($N_p = 5$, $N_g = 1$) |
| Bars (Fig. 11) | 6-8-9 | 0.0001 | 0.05 | 20 | 32 | 32 | rand-cross-randflip ($N_p = 5$, $N_g = 1$) |
| Bars (Fig. 11) | 8-8-16 | 0.0001 | 0.05 | 20 | 32 | 64 | *random sampling* ($N_{\text{new}} = 20$, $p(z_h = 1) = \frac{1}{H_0}$) |
| Bars (Fig. 11) | 8-8-16 | 0.0001 | 0.05 | 20 | 32 | 64 | fit-randflip ($N_p = 5$, $N_c = 4$, $N_g = 1$) |
| Bars (Fig. 11) | 8-8-16 | 0.0001 | 0.05 | 20 | 32 | 64 | rand-randflip ($N_p = 5$, $N_c = 4$, $N_g = 1$) |
| Bars (Fig. 11) | 8-8-16 | 0.0001 | 0.05 | 20 | 32 | 64 | fit-cross-randflip ($N_p = 5$, $N_g = 1$) |
| Bars (Fig. 11) | 8-8-16 | 0.0001 | 0.05 | 20 | 32 | 64 | rand-cross-randflip ($N_p = 5$, $N_g = 1$) |
| van Hateren (Fig. 12) | 300-300-256 | 0.0001 | 0.001 | 10 | 32 | 100 | fit-randflip ($N_p = 8$, $N_c = 7$, $N_g = 2$) |
| House (Fig. 3D,E$^{\dagger}$) | 64-64-64 | 0.0001 | 0.01 | 20 | 32 | 200 | fit-randflip ($N_p = 10$, $N_c = 9$, $N_g = 4$) |
| House (Fig. 3E$^{\ddagger}$) | 512-512-144 | 0.0001 | 0.05 | 20 | 32 | 64 | fit-randflip ($N_p = 5$, $N_c = 4$, $N_g = 1$) |
| Barbara (Fig. 4) | 50-500-500-64 | 0.0001 | 0.001 | 160 | 512 | 100 | fit-randflip ($N_p = 5$, $N_c = 4$, $N_g = 1$) |
| Barbara (Fig. 4) | 256-256-512-64 | 0.0001 | 0.001 | 160 | 512 | 100 | fit-randflip ($N_p = 5$, $N_c = 4$, $N_g = 1$) |
| House 50 (Fig. 5) | 512-512-144 | 0.0001 | 0.01 | 20 | 32 | 64 | fit-randflip ($N_p = 5$, $N_c = 4$, $N_g = 1$) |
| Castle 50 (Fig. 5) | 512-512-25 | 0.0001 | 0.00125 | 20 | 32 | 32 | fit-randflip ($N_p = 5$, $N_c = 4$, $N_g = 1$) |
| Castle 80 (Fig. 5) | 512-512-144 | 0.0001 | 0.001 | 20 | 32 | 64 | fit-randflip ($N_p = 5$, $N_c = 4$, $N_g = 1$) |
| CIFAR-10 (Fig. 15) | 32-512-3072 | 0.0004 | 0.009 | 160 | 128 | 20 | fit-randflip ($N_p = 15$, $N_c = 10$, $N_g = 1$) |
| CIFAR-10 (Fig. 15) | 50-500-500-3072 | 0.0004 | 0.009 | 160 | 128 | 20 | fit-randflip ($N_p = 15$, $N_c = 10$, $N_g = 1$) |
| CIFAR-10 (Fig. 15) | 1024-256-512-3072 | 0.0004 | 0.009 | 160 | 128 | 20 | fit-randflip ($N_p = 15$, $N_c = 10$, $N_g = 1$) |
| CIFAR-10 (Fig. 15) | 1024-512-3072 | 0.0004 | 0.009 | 160 | 128 | 20 | fit-randflip ($N_p = 15$, $N_c = 10$, $N_g = 1$) |
| Audio (Fig. 16) | 200-512-400 | 0.0001 | 0.001 | 160 | 512 | 20 | fit-randflip ($N_p = 15$, $N_c = 1$, $N_g = 1$) |

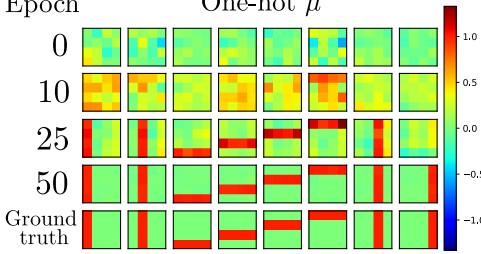

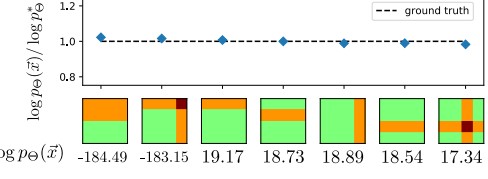

Figure 8: TVAE Training on Simple Bars Data: Noiseless Output of the TVAE's DNN for the 8 Possible One-hot Input Vectors Over Several Training Epochs. Generating parameters are in the last row.

Figure 9: Correlated Bars Test. The plot shows the ratio between inferred and ground-truth log-likelihoods $\log p_{\Theta}(\vec{x})$ of data points with interesting bar combinations. The inferred values are reported below the data points themselves.

section of the figure shows the ELBO values (averages over all batches for each epoch) as training progresses. The cyclic learning rate schedule is responsible for the oscillatory behavior. The right section shows some example data points together with samples from the trained TVAE model that reached the highest ELBO value out of the ten runs.

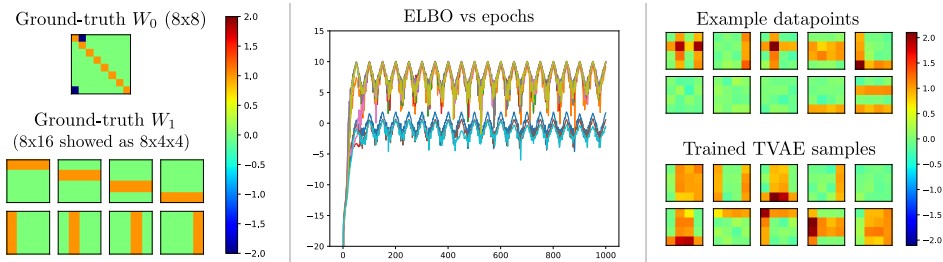

Figure 10: Generative Parameters for the Correlated Bars Test (left); ELBO Values over Epochs for 10 runs (center); Example Datapoints and Samples from the Generative Model (right).

Using bars test data generated with a linear model (no correlations between pairs of bars) as described above, Fig. 11 depicts reliabilities of different evolutionary operator designs (also compare App. A.1).

### B.3    SCALABILITY EXPERIMENTS

The $N = 100,000$ image patches of size $D = 16 \times 16$ for the scalability experiments were extracted from a standard image database (van Hateren & van der Schaaf, 1998) and pre-processed as in Guiraud et al. (2018). The most elementary VAEs use a linear mapping for the decoder $\vec{\mu}(\vec{z}, W)$. While being a natural baseline, such VAEs have also played a role in understanding important properties of VAE learning (e.g. Dai et al., 2018; Lucas et al., 2019). For standard Gaussian latents, a linear VAE can be shown to recover probabilistic PCA solutions. For Bernoulli latents, we can recover binary sparse coding (Haft et al., 2004; Shelton et al., 2011) in case of linear decoders. We therefore began our analysis (using $H = 300$ latents) with a linear VAE. After 100 epochs the weights of the linear mapping were used to initialize the bottom layer of a deeper decoder network with three layers of 300, 300 and $16 \times 16 = 256$ units, respectively. The weights of the deeper layers were simply initialized to the identity matrix. Furthermore, prior and variance were optimized. The setup just described guarantees a common starting point for linear and non-linear VAEs such that the difference provided by deeper decoder DNNs can be highlighted. Fig. 12 shows the variational bounds during learning of the linear VAE compared to the non-linear VAE for a typical experiment. The non-linear VAE can be observed to quickly and significantly optimize the lower bound beyond a linear VAE. In proceeding experiments (when we are not concerned with comparisons to linear VAEs) we simply optimized the weights of the non-linear TVAE directly as we did not observe an advantage in first optimizing a linear VAE.

Regarding scalability, we observed a similar efficiency of non-linear TVAE compared to linear models as described in the main text. To further investigate scalability, we went to TVAEs with up to $H$=1000 latent variables (while using 100 units in the DNN middle layer). TVAE training time remained in line with the theoretical linear scaling with $H$ while the variational bound further increased.

### B.4    IMAGE DENOISING

*Data Estimator.* Given a trained TVAE with parameters $\Theta$, we estimated the value of a pixel in a single patch as $x_d^{\text{est}} = \mathbb{E}_{p_\Theta(x_d|\vec{x})}[x_d]$. When using $p_\Theta(x_d \mid \vec{x}) = \sum_{\{\vec{z}\}} p_\Theta(x_d \mid \vec{z}) p_\Theta(\vec{z} \mid \vec{x})$ we obtain:

$$x_d^{\text{est}} = \mathbb{E}_{p_\Theta(\vec{z}|\vec{x})}\left[\mathbb{E}_{p_\Theta(x_d|\vec{z})}[x_d]\right] = \mathbb{E}_{p_\Theta(\vec{z}|\vec{x})}[\mu_d(\vec{z})]. \quad (15)$$

The expectation value on the right-hand-side of Eqn. (15) is then approximated based on the encoding parameters $\Phi^{(n)}$ using truncated posteriors. Finally, we took a weighted average of the estimates

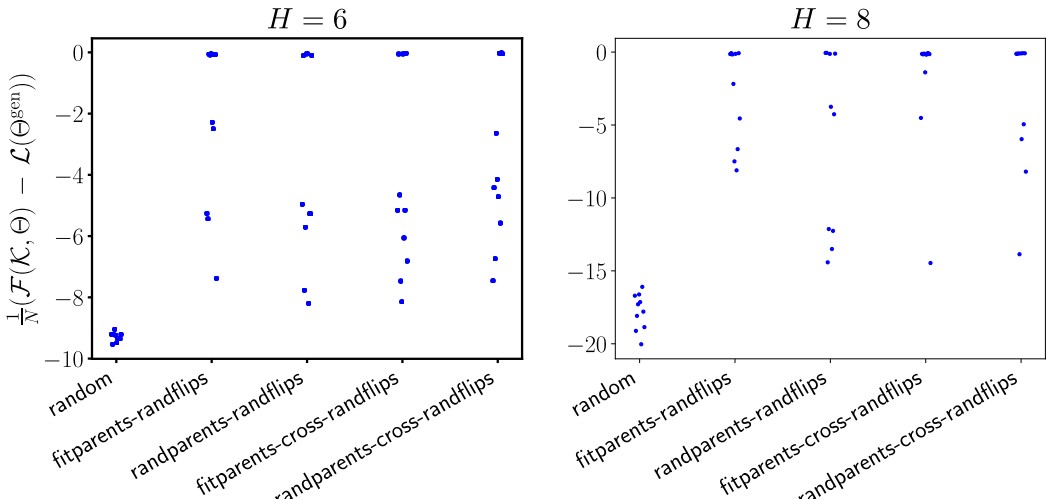

Figure 11: Reliability of different evolutionary operator combinations (cf. App. A.1) on bars test (see App. B.2 for details). Reliability is quantified in terms of the normalized difference between the lower bound at the last training epoch and the log-likelihood computed using the ground-truth generating parameters. For the strategy labeled as 'random', we randomly sampled states from a Bernoulli distribution with $p(z_h = 1) = \frac{1}{H}$.

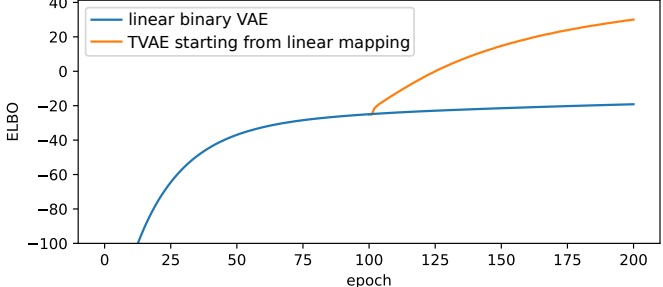

Figure 12: ELBO Gain of TVAE Compared to Linear VAE with Binary Latents (on $16 \times 16$ image patches).

of a pixel value in different patches (see, e.g., Burger et al., 2012) in order to generate the pixel values of the full denoised image.

*Performance Comparison Under Controlled Conditions.* To evaluate the performance on standard denoising benchmarks, we first compared TVAE to related probabilistic sparse coding approaches such as MTMKL, GSC and var-BSC (Fig. 3 D). MTMKL and GSC use the data model of spike-and-slab sparse coding and for training mean-field and truncated posterior approximations with pre-selection are used, respectively. Compared to MTMKL and GSC, var-BSC uses a less complex data model and a training scheme also based on evolutionary optimization (Guiraud et al., 2018). The denoising performance observed in the scenario with controlled conditions (Fig. 3 D) shows that for high noise level ($\sigma = 50$), var-BSC achieves higher PSNR values than MTMKL and GSC although the method uses a simpler data model. This observation demonstrates the effectiveness of the evolutionary training method used by var-BSC. However, PSNR values for TVAE are significantly higher due to the higher flexibility in modeling the data distribution provided by the used DNN.

*Performance Comparison Under Uncontrolled Conditions.* In a second step, Fig. 3 E compared the performance of TVAE with respect to different denoising approaches including deterministic sparse coding (KSVD), a mixture model approach (EPLL), a non-local image processing method (WNNM) and state-of-the-art denoising methods based on deep neural networks (BDGAN and DPDNN). These approaches can be distinguished, e.g., by the amount of employed training data and by the requirement for clean data. TVAE as well as MTMKL, GSC and var-BSC do not require clean images for training. Furthermore, all these approaches can be trained if only the single noisy image is available ('zero-shot' learning; compare, e.g., Shocher et al., 2018; Imamura et al., 2019). Instead, EPLL, BDGAN and DPDNN use clean training data (typically tens or hundreds of thousands of data points are collected for training).

Approaches such as noise2noise (n2n Lehtinen et al., 2018) and noise2void (n2v Krull et al., 2019b) occupy a middle ground: they can be trained on noisy data but they typically require much larger amounts of data than, e.g., TVAE or MTMKL. In the original n2v publication, for instance, 400 (noisy) $180 \times 180$ BSD (Martin et al., 2001) images were used to create a training dataset (this procedure also involved data augmentation; compare Krull et al. 2019b). For our comparison with results of Fig. 3 E, we used the standard, publicly available code for n2v (Krull et al., 2019a) together with the default training set ($\sigma = 25$) employed in the original n2v publication (specifically, we followed the 'denoising2D_BSD68' example from the corresponding GitHub repository). We then applied the trained n2v network to denoise the 'house' image with $\sigma = 25$. The resulting PSNR value was $32.10dB$ which is $0.76dB$ lower than the PSNR value for BM3D ($32.86dB$). The difference is consistent with an on average $0.88dB$ lower performance of n2v compared to BM3D on the BSD68 test set (see Krull et al., 2019b). The same network can also be used to denoise an image with lower or higher noise level. The n2v network trained on $\sigma = 25$ does, for instance, result in PSNR values of $32.93dB$ for the 'house' image with $\sigma = 15$ and in $20.96dB$ for the 'house' image with $\sigma = 50$ (see $n2v^\dagger$ in Tab. 2). Especially for high noise levels performance can be much improved, however, if the n2v network is trained using images with the same noise level as the test image. In order to do so, we followed the procedure described in the n2v publication while adapting the noise level of $\sigma = 15$ in one case and $\sigma = 50$ for the other case (as described above, we used the 'denoising2D_BSD68' example as reference implementation). Trained on a dataset with matched noise, we then denoised the 'house' image with $\sigma = 15$ in the one, and $\sigma = 50$ in the other case (results listed as n2v$^\ddagger$ in Tab. 2). The PSNR values obtained for 'house' in this matched-noise-level scenario are much higher compared to the scenario with unmatched noise level (e.g., for $\sigma = 50$ the PSNR improvement is approximately 8 dB). The much lower performance for mismatched noise for n2v is in this respect consistent with observations for standard DNN denoising for which training with the ground-truth noise level has been pointed out as important for performance (Chaudhury & Roy, 2017; Zhang et al., 2018).

The n2v approach can avoid having to know the exact noise level, e.g., if it is trained on just the single noisy image. In a last experiment, we hence investigated this 'zero-shot' denoising feature of n2v and applied the algorithm to denoise the 'house' image while using the same noisy image for training that we seek to denoise (again, our implementation was based on the n2v 'denoising2D_BSD68' example, see above). The obtained PSNR values are listed as n2v* in Tab. 2.

From Tab. 2 it can be observed, that for all considered training settings of n2v and all noise levels, PSNR values of TVAE are consistently higher than those of n2v even if n2v is trained on external

Table 2: Denoising Performance of n2v in PSNR (dB) for the 'House' Image. For comparison, we additionally list the performance of TVAE (numbers copied from Fig. 3 E). PSNR values for n2v$^\star$ are obtained by training only on the noisy image (i.e., in the same setting as used for MTMKL, GSC, var-BSC and TVAE in Fig. 3 E). More training data improves performance for n2v. PSNR values for n2v$^\dagger$ show performance if additional training data in the form of noisy images with AWG noise $\sigma = 25$ is used. Further improvements (especially for high noise) are obtained if the n2v network is trained on training data with a noise level that matches the noise of the test set (see n2v$^\ddagger$). For instance, we used for n2v$^\ddagger$ training data with $\sigma = 50$ to denoise the 'house' with $\sigma = 50$. PSNR values were computed using the model in its state at the training epoch with smallest validation loss; for n2v$^\star$, we performed three independent runs of the algorithm and here report the results of the best run (in terms of validation loss). See text for further details.

|  | $\sigma$=15 | $\sigma$=25 | $\sigma$=50 |
|---|---|---|---|
| n2v$^\star$ | 32.22 | 29.69 | 24.89 |
| n2v$^\dagger$ | 32.93 | 32.10 | 20.96 |
| n2v$^\ddagger$ | 33.91 | 32.10 | 28.94 |
| TVAE | **34.27 ± .02** | **32.65 ± .06** | **29.98 ± .05** |

data with matched-noise level. Additional parameter tuning may improve performance of n2v$^*$ to a certain extent but PSNRs are in general much lower than n2v$^\ddagger$. While we followed for n2v$^\ddagger$ the standard hyperparameter setting of the original paper/code publication of n2v (Krull et al., 2019b;a), we cannot exclude further improvements with parameter fine tuning for the 'house' benchmark. However, we remark that the difference of n2v$^\ddagger$ and BM3D for the 'house' benchmark is on the very same range as the differences between n2v and BM3D as reported on the BSD data set in the original n2v publication. The stronger performing BM3D is according to denoising performance the preferable comparison and as such included in Fig. 3 E. In terms of efficiency, the n2v approach is in general (once trained) faster than BM3D as well as TVAE, however.

PSNR values of noise2noise (n2n) are usually very closely aligned with PSNR values achievable by feed-forward DNNs. More concretely, n2n uses, for instance, a RED30 network (Mao et al., 2016) which achieves 31.07 dB PSNR on the BSD300 data set if trained on clean data. If directly trained on noisy data, RED30 achieves 31.06 dB (Lehtinen et al., 2018). n2n is thus strongly performing in terms of PSNR. The caveat of n2n compared to n2v is, however, that the noisy data n2n uses is rather artificial. The pairs of images n2n is trained on consist of two different noise realization of the same underlying clean image. For real data, such a setting is only approximately occurring at most, which has motivated the n2v approach.

Like n2v, BDGAN and DPDNN are optimized for specific noise levels (specific standard deviations are used to generate the noisy training examples). EPLL is trained exclusively on clean image patches; for denoising, the algorithm requires the ground-truth noise level of the test image as input parameter. Ground-truth noise level information is also required by KSVD and WNNM.

Like all approaches in the top category of Fig. 3 E, TVAE does not require ground-truth noise level information, nor clean images, nor large amounts of training data. For the 'zero-shot' setting, TVAE is consequently the best performing system on the 'house' benchmark. Such a high performance is notably achieved using a basic DNN and relatively small patch sizes of $D = 8 \times 8$ (for $\sigma = 15$ and $\sigma = 25$) or $D = 12 \times 12$ (for $\sigma = 50$). All feed-forward DNNs for denoising use much larger patches (e.g., n2v use $64 \times 64$). That a competitive denoising performance can be achieved for small patches, in general, argues in favor for VAE approaches to denoising. Indeed, TVAE even comes close to state-of-the-art approaches (BDGAN and DPDNN) that use very intricate DNN architectures and large amounts of clean training data. We believe that such results underline the potential of the here investigated approach although the novelty of the approach is the focus rather than extensive benchmarking.

*Comparison to Deep Generative Models.* While denoising is, in general, well suited for deep generative models, performance for standard image denoising is not as common as such results for standard DNNs (which may also be related to efficiency aspects). Exceptions include the recent BDGAN (Zhu et al., 2019) and the very recent DivNoising (Prakash et al., 2021) and NN+X (Zheng

et al., 2021) approach. Regarding VAEs, we here compared TVAE to GSVAE (Jang et al., 2016) and VLAE (Park et al., 2019a) using publicly available source code of these models (Jang (2016) and Park et al. (2019b), respectively). For GSVAE, we implemented continuous latents, i.e. we used a Gaussian distribution over the observables (rather than a Bernoulli as implemented in the public source code). For both GSVAE and VLAE, we used the same hyperparameters as in the respective publicly available implementations. To apply both models for denoising, we used the same patch-based processing as employed for TVAE (compare *Data Estimator* paragraph above). The PSNRs listed in Fig. 3 were obtained using patch sizes of $12 \times 12$ and $8 \times 8$ for GSVAE and VLAE, respectively, and they were measured at the training epoch with the highest lower bound.

*Computational Demand.* An important limitation of TVAE is its computational demand. For our experiments on the 'house' image with noise level $\sigma = 50$ in Fig. 3 E we used $N = 60025$ patches of $D = 12 \times 12$ pixels, which amounts to all possible non-overlapping square patches of that size that can be extracted from the image. For training and denoising we used a TVAE with $H = 512$ latent variables, sizes of $|\Phi^{(n)}| = 64$, and 512 units in the DNN middle layer of the decoder. TVAE training required 49 seconds per training epoch when executing on a single NVIDIA Titan Xp GPU and $2.5$ GB of GPU memory. We ran for $500$ epochs which required between seven and eight hours on the single GPU. We did not observe significant changes in variational bound values or in denoising performance after 500 epochs in any of the experiments we conducted for Fig. 3. Runtime complexity increased linear with the number of data points $N$, with the dimensionality of the data $D$, with the number of the latents $H$, and with the size of the DNN used. Runtimes also increased approximately proportional w.r.t. the size of $\Phi^{(n)}$. Empirically we observed a sub-linear scaling with $|\Phi^{(n)}|$ presumably because of significant overhead computations: for example, increasing from $|\Phi^{(n)}| = 64$ to $|\Phi^{(n)}| = 128$ (while keeping all other parameters as above) computational time increases from 49 seconds per training epoch to 75 seconds.

For noise levels $\sigma = 15$ and $\sigma = 25$ in Fig. 3 E we used smaller patch sizes ($D = 8 \times 8$) and fewer stochastic latents ($H = 64$) but larger $\Phi^{(n)}$ (i.e., $|\Phi^{(n)}| = 200$). In general, if the patch size $D$ is increased, more structure has to be captured. This can be done either by increasing the size of the stochastic latents $H$ or by using larger DNNs. Both, in turn, requires more training data in order to estimate the increased number of parameters. In the current setup, the sizes of $D$ which are currently feasible are comparably small. The denoising performance based on small patches is, however, notably very high.

For comparison, n2v uses up to $D = 64 \times 64$ and also all other feed-forward DNN approaches use significantly larger patch sizes than TVAE (and the other approaches in category 1). Still, n2v can be trained efficiently on large patches requiring approximately 19 hours on a NVIDIA Tesla K80 GPU for training on approximately 3k noisy images of shape 180x180 and seconds for the denoising of one 256x256 image. The higher computational demand of TVAE is also the reason why averaging across databases with many images (such as BSD68) or applications to large single images quickly becomes infeasible. As a novel approach, TVAE is, however, far from being fully optimized algorithmically compared to large feed-forward approaches, and there is certainly further potential to improve training efficiency.

Tab. 3, furthermore, systematically compares runtimes of GSVAE, VLAE and TVAE for the Barbara denoising benchmark from Fig. 4.

### B.5 ZERO-SHOT INPAINTING

Similarly to the use of TVAE for denoising (see B.4), a single image with missing pixels (Fig. 13, center) is divided into square patches to form the training set; during training, missing pixels can be treated as observables with unknown values when evaluating log-joint probabilities of a data-point, which provides a grounded way to treat missing data. To estimate likely values of missing pixels, we again use the estimator from (15) and only consider the observed entries of a data point when evaluating posterior probabilities:

$$x_d^{\text{est}} = \mathbb{E}_{p_\Theta(\vec{z}|\vec{x}^{\text{obs}})}\big[\mu_d(\vec{z})\big] . \tag{16}$$

As for denoising (compare Eqn. 15), the expectation value can be approximated using truncated posteriors with encoding parameters $\Phi^{(n)}$. Complete data points with 'inpainted' pixels can then

Table 3: Runtimes required by GSVAE, VLAE and TVAE to reach the lower bounds and PSNRs reported in Fig. 4B. † Total runtime of VLAE consists of 5.1 hrs for parameter optimization and 1.8hrs for importance sampling. The PyTorch implementations of the models were executed on a single NVIDIA Tesla V100 NV-Link 32GB HBM2 on a server with Intel Xeon 4214 12-core 2.20 GHz CPUs.

| GSVAE | | | | VLAE | | TVAE | | | |
|---|---|---|---|---|---|---|---|---|---|
| 50-500-500-64 | | 256-256-512-64 | | 50-500-500-64 | | 50-500-500-64 | | 256-256-512-64 | |
| epoch | total | epoch | total | epoch | total | epoch | total | epoch | total |
| 10s | 0.01hrs | 11s | 0.03hrs | 46s | 6.9†hrs | 213s | 9.5hrs | 234s | 10.4hrs |

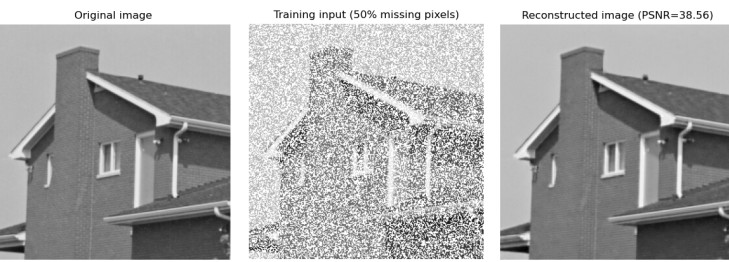

Figure 13: Inpainting of the 'House' Image with TVAE.

be used for DNN backpropagation to optimize the decoder. The inpainting procedure consequently directly derives from the standard probabilistic treatment of missing values. In contrast, amortized approaches will have to specify how an encoder network should treat missing values because encoder DNNs expect real values for all pixels as input.

When evaluating TVAE on standard inpainting benchmarks, we observe competitive performance compared to other approaches. Fig. 5 shows a comparison of inpainting performance (in terms of PSNR) with previous state-of-the-art systems that like TVAE do not require large, clean training data nor information, e.g., on the noise level. As can be observed, TVAE outperforms approaches such as BPFA (Zhou et al., 2012) or Papyan et al. (2017) in most settings. TVAE performance is lower than for DIP (Ulyanov et al., 2018). TVAE, like BPFA and Papyan et al. (2017), is a permutation-invariant approach, however. That is, the TVAE model itself does not leverage information about the 2D nature of images. In contrast, DIP results rely on a large dedicated DNN with LeakyReLU as activation functions, a U-net / hourglass architecture with skip connections, and convolutional units with reflection padding (see supplement of Ulyanov et al. (2018)). The convolutional stages do explicitly assume the 2D image structure. We also remark that DIP uses in total 2 million parameters compared to about 0.5 million parameters of the standard multi-layer perceptron used in TVAE. Furthermore, and as a consequence of its more intricate architecture, DIP uses many more tunable hyper-parameters.

In the case of the 'House' image, training lasted 500 epochs taking around 60 seconds/epoch on a single NVIDIA Titan Xp GPU.

Tab. 1 lists the hyper-parameters used for these experiments.

### B.6    FURTHER COMPARISON TO MODELS WITH GRADIENT-BASED ENCODER OPTIMIZATION

Fig. 4 provides a comparison of the generative representations used by TVAE, GSVAE (Jang et al., 2016) and VLAE (Park et al., 2019a) for denoising $D = 8 \times 8$ patches of the noisy Barbara image (additive white Gaussian noise with $\sigma = 25$; compare Sec. 3 and App. B.4). We applied the models using different decoder architectures including $50 - 500 - 500 - D$ and $256 - 256 - 512 - D$ (denoted are the numbers of network units counted from decoder input layer (value referred to as

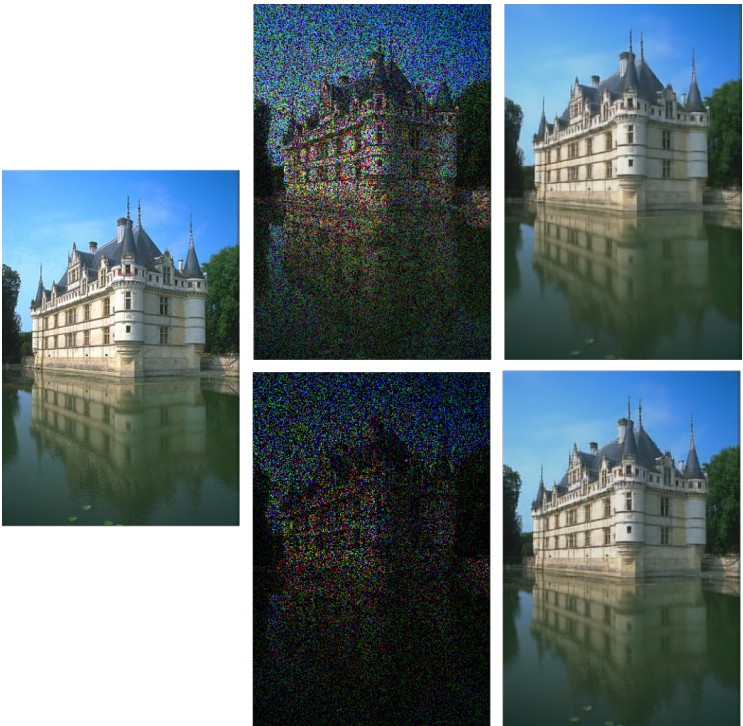

Figure 14: Inpainting of the 'Castle' Image with TVAE. **Left** Original image. **Center** Training image (top, 50% missing pixels, bottom, 80% missing pixels). **Right** Inpainting result with TVAE (top, 50% missing pixels, bottom, 80% missing pixels).

$H$), to decoder hidden layers, to decoder output layer (value referred to as $D$); for GSVAE we used $H = 5 \times 10$ and $H = 32 \times 8$ categorical latents). We observed the GSVAE and VLAE implementations to yield significantly better results when applying the algorithms after normalizing the image patches to the interval [0, 1]; to maintain comparability, we applied the same procedure for TVAE.

To investigate differences in the learned representation, we computed (i) the sparsity of the encoding ($\sum_h \pi_h / H$; i.e. the average number of active latents per data point), (ii) the best lower bound across training epochs (both depicted in Fig. 4B), and (iii) the decoder output for singleton input ($H$-dimensional vector containing a single '1' entry and '0' entries otherwise; results visualized in Fig. 4C). For VLAE, Fig. 4B lists an importance sampling-based log-likelihood estimate instead of a lower bound value. The results depicted in Fig. 4A and C were obtained using the $50-500-500-D$ decoder, and panel C illustrates a random selection of ten singleton means. As can be observed from Fig. 4, TVAE learns comparably very sparse encodings with generative fields capturing distinct patterns of the underlying image. Such encoding shows to result in comparably very high lower bounds which, in turn, result in comparably very high denoising PSNRs.

As another example of a commonly used image patch dataset, we applied TVAE to CIFAR-10 and investigated the learned data encoding, also comparing with GSVAE and VLAE (Fig. 15). While patches from the Barbara image (considered above) contain comparably little, elementary structure, CIFAR-10 images depict entire objects (cars, animals, etc.) and thus contain significantly more complex structure. Similarly to the Barbara benchmark, TVAE learns a sparse encoding on CIFAR-10 images; the performance in terms of lower bounds is, however, less competitive compared to GSVAE and VLAE (when controlling for comparable decoder network architectures).

**A** Log-likelihood estimates

| GSVAE | VLAE | | TVAE | | | |
|---|---|---|---|---|---|---|
| 1024-256-512 | 32-512 | 50-500-500 | 32-512 | 50-500-500 | 1024-256-512 | 1024-512 |
| 2559 | 2392* | 2687* | 1649 | 1831 | 2044 | 2893 |

**B** Sparsity

| GSVAE | VLAE | | TVAE | | | |
|---|---|---|---|---|---|---|
| 1024-256-512 | 32-512 | 50-500-500 | 32-512 | 50-500-500 | 1024-256-512 | 1024-512 |
| $\frac{512}{1024}$ | n/a | n/a | $\frac{13}{32}$ | $\frac{18}{50}$ | $\frac{179}{1024}$ | $\frac{135}{1024}$ |

**C** Samples (top) and TVAE reconstructions (bottom)

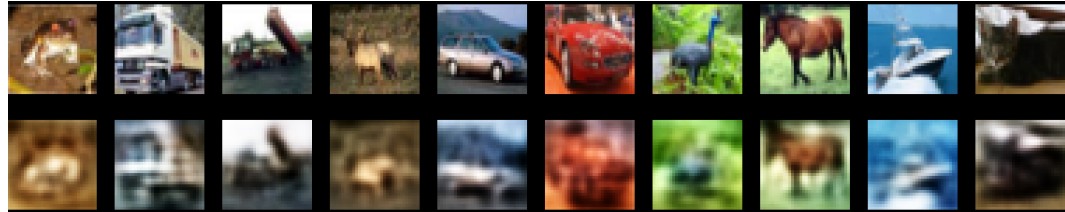

Figure 15: Log-likelihood estimates and sparsity obtained for CIFAR-10. For comparison, we considered Gumbel-softmax VAE (GSVAE; Jang et al., 2016)) and VLAE (Park et al., 2019a). For VLAE, we cite the values reported in the original publication (these correspond importance sampling-based log-likelihood estimates). Values for GSVAE, we computed ourselves using publicly available source code (Jang, 2016) together with the default parameter settings. For GSVAE and TVAE, the values listed in panel A denote lower bounds. The values in panel B depict sparsity measured in terms of average number of active latents per data point ($\sum_h \pi_h / H$). The columns in panel A, B refer to different decoder architectures: 1024-256-512, for instance, denotes $H = 1024$ latents in the decoder input layer and two hidden layers with 256 and 512 hidden units, respectively. The decoder output layer was the same in all conditions ($D = 32 \times 32 \times 3$). Panel C depicts reconstructed data points obtained with TVAE for the 1024-512 setting.

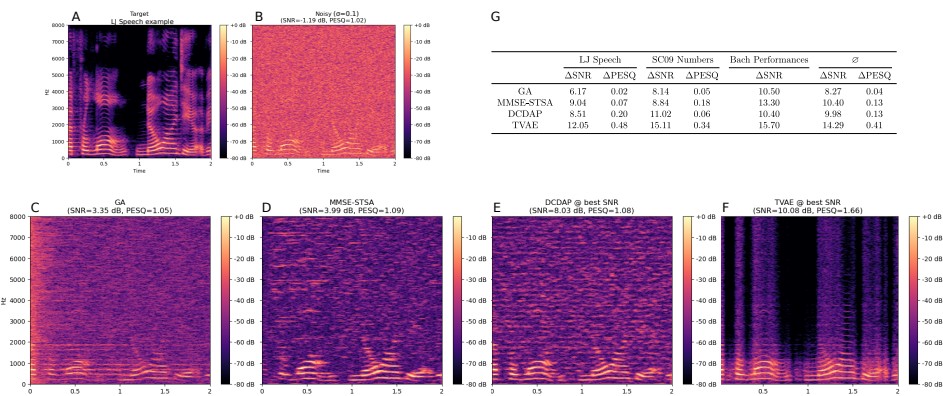

Figure 16: Results for 'Zero-Shot' Audio Denoising. Panel A depicts the dB-scaled amplitude spectrogram of an example from the LJ-Speech dataset. Panel B was obtained after adding Gaussian noise ($\sigma = 0.1$). Panels C-F show the denoising results of GA (Lu & Loizou, 2008), MMSE-STSA (Ephraim & Malah, 1984), DCDAP (Narayanaswamy et al., 2021b) and TVAE. Panel G reports SNR and PESQ improvements averaged over five examples per dataset (the improvement is computed by comparing the denoised waveform with the target waveform; see App. B.7 for details).

## B.7 ZERO-SHOT AUDIO DENOISING

Learning-based 'zero-shot' data enhancement is subject of current research not only in the context of image but also audio processing (e.g., Zhang et al., 2020; Narayanaswamy et al., 2021b; Michelashvili & Wolf, 2020). Narayanaswamy et al. (2021b), for instance, discuss applications of U-Net-based networks with dilated convolutions to the task of unsupervised Gaussian audio denoising. To address the question of the extent to which the strong 'zero-shot' denoising performance of TVAE on visual data (Figs. 3, 4) carries over to audio data, we applied TVAE to the benchmark considered in Sec. 4.1 and Tab. 1 in Narayanaswamy et al. (2021b). Concretely, we followed the experimental design described in Narayanaswamy et al. (2021b) and used the following datasets: LJ-Speech (Ito & Johnson, 2021), SC09 Spoken Numbers (Warden, 2018; Donahue et al., 2018; we used the version made available by Bhalley, 2018), and Bach Performances (Donahue et al., 2018). From each of these three datasets, we randomly selected five examples after resampling the audio files to 16 kHz. Regarding the number of examples per dataset, our procedure slightly diverts from the one used by Narayanaswamy et al. (2021b) who collected 50 examples per dataset. The reason for limiting our evaluation to only five samples lies in avoiding long runtimes of TVAE (compare App. A and B.4). For the LJ-Speech and Bach Performances datasets, we randomly selected excerpts of two seconds duration; for SC09, each example had a duration of one second. To each of the in total 15 collected snippets, we added Gaussian noise using $\sigma = 0.1$. We then normalized the noisy waveforms to fill the range [-1, 1]. Narayanaswamy et al. (2021b) do not report to have used such normalization; for TVAE, we observed data normalization to be beneficial as it helped to avoid numerical instabilities related to very small variances learned due to very small waveform amplitudes. Finally, we applied TVAE to the noisy waveform after extracting overlapping chunks of $D = 400$ samples (similarly to the overlapping patches extracted from noisy images, cf. App. B.4).

For the evaluation, we used two standard baselines, namely a Geometric Approach (GA; Lu & Loizou, 2008) and the Minimum-Mean Square Error Short-Time Spectral Amplitude estimator (MMSE-STSA; Ephraim & Malah, 1984) together with the noise power spectral density estimator of Gerkmann & Hendriks (2012). Furthermore, we compared to the approach of Narayanaswamy et al. (2021b) which we refer to as DCDAP (as an abbreviation for 'Deep Audio Prior with dilated convolutions '). We applied the publicly available source code of DCDAP (Narayanaswamy et al., 2021a) together with the default parameter settings; as 'dilation type', we chose the option 'exponential'. For the LJ-Speech and the SC09 datasets, we quantified performance in terms of Signal-to-Noise Ratio (SNR) and Perceptual Evaluation of Speech Quality (PESQ); for the non-speech dataset (Bach Performances), we measured only SNRs. For all three benchmark datasets considered, TVAE can improve the performance of all compared methods in terms of both SNR and PESQ improvement (Fig. 16).

