# OpenReview forum: "Direct Evolutionary Optimization of Variational Autoencoders With Binary Latents"
_ICLR.cc/2022/Conference — ICLR 2022 Submitted_

### Official Review · Reviewer_DBE8 · 2021-10-26

**Correctness:** 3
**Technical Novelty And Significance:** 2
**Empirical Novelty And Significance:** 2
**Recommendation:** 5
**Confidence:** 4

**Main Review:**

Strengths:
- The proposed idea is rather straightforward but interesting. To my knowledge, it is the first time I see an application of an evolutionary algorithm to learn binary encoders in VAEs.
- The experiments are promising. However, I miss a deeper comparison with existing methods (e.g., RELAX, Gumbel-softmax).

Weaknesses:
- It is quite unclear how the decoder is formulated. The Eq. 3 suggests we deal with the Gaussian distribution. However, the comment on the MSE is a bit misleading because it is a well-known fact that the likelihood function for the Gaussian distribution results in the MSE.
- The notion of the subsets (\Phi^{(i)}) that are variational parameters is a bit vague. The paper would highly benefit from presenting an example right after the Eq. 3. This is especially important in the discussion about passing all states of \Phi^(n)} through the decoder and the complexity.
- In the Eq. 5 (and before it), there is a new variable called “prior parameters” introduced. However, the authors did not introduce it earlier nor define the prior. From the equation it seems they are the expected values of z’s.
- What is the rationale for picking the very particular mutation operator? What about other possible perturbations?
- I miss a deeper comparison and discussion with other approaches, namely:
    - MCMC techniques for binary variables:
        * Strens, M. (2003). Evolutionary MCMC sampling and optimization in discrete spaces. In Proceedings of the 20th International Conference on Machine Learning (ICML-03) (pp. 736-743).
        *Auzina, I. A., & Tomczak, J. M. (2021). Approximate bayesian computation for discrete spaces. Entropy, 23(3), 312.
    - Stochastic differentiation libraries that allow differentiation through discrete random variables:
        * Bingham, E., Chen, J. P., Jankowiak, M., Obermeyer, F., Pradhan, N., Karaletsos, T., ... & Goodman, N. D. (2019). Pyro: Deep universal probabilistic programming. The Journal of Machine Learning Research, 20(1), 973-978.
        * van Krieken, E., Tomczak, J. M., & Teije, A. T. (2021). Storchastic: A Framework for General Stochastic Automatic Differentiation. arXiv preprint arXiv:2104.00428.
    - Or methods for differentiating black-box models:
        * Grathwohl, W., Choi, D., Wu, Y., Roeder, G., & Duvenaud, D. (2017). Backpropagation through the void: Optimizing control variates for black-box gradient estimation. arXiv preprint arXiv:1711.00123.
- The structure of the network is very simplified. It is hard to predict how the proposed approach will behave in a more complex case (i.e., convolutional layers, recurrent layers, residual layers, etc.). I am aware that the proposed approach is generic, however, the learning dynamics is completely different for various layers.


**Summary Of The Paper:**

The paper proposes to use evolutionary computing for optimizing binary encoders in the Variational Auto-Encoder framework. The authors present a specific evolutionary algorithm for learning encoders for binary latent variables. The experimental section outlines some applications to denoising and inpainting.

**Summary Of The Review:**

The paper proposes an evolutionary algorithm for learning encoders for binary latent variables. The paper allows a reader to understand the main idea, however, it lacks a proper presentation of some details (e.g., the idea of using truncated distributions in the discussed context, presenting explicitly the forms of distributions used). Moreover, the experimental section is interesting, however, it misses a deeper comparison with other existing methods and an analysis of various architectures of neural networks. Overall, I find the paper interesting, however, it is not ready to be accepted to the conference.

---

> ### Author Response · Authors · 2021-11-23
> **Reply to points raised**
>
> Points 1 & 2 (Strength): Yes, indeed we were interested in a straightforward/direct way to optimize discrete VAEs. The very different and weaker assumptions on the encoder allow us to study effects of amortized DNN-based approaches (+ sampling approximation etc) followed by more conventional approaches. We do consider the approach itself as well as the gained insights into representations as important (please see general reply).
>
> Point 3 & 4: We use a Gaussian in Eq. 1, which later translates to squared error in Eq. 7 & 8, true. We kept the notation usually general in between as the approach does not rely on a Gaussian assumption / mean-square error. We made some general changes to clarify, thanks for pointing out.
>
> Point 5: We now explicitly introduce the prior parameters $\pi_h$ after Eq. 1, sorry and thanks for pointing this out.
>
> Point 6: We have experimented with different combinations of evolutionary operators. The chosen combination 'fitparents-randflips' offered the best trade-off between effectiveness and efficiency (compare Fig. 11). We now comment in the text, thanks.
>
> Point 7: Thanks for the pointers to references. Please do see our general reply for this point, and the there described changes to our manuscript.
>
> Point 8: Our focus in ongoing work remains the study of the properties of learned representations for different data, and we do investigate other than Gaussian noise, for instance. Further improvements (both technical/algorithmical and methodological) will allow for increasingly large/intricate DNNs in the future, which will indeed be interesting; and indeed the approach can be applied without modifications to VAEs with more complex networks (see, e.g.,  Eq. 9 and comparison Eqs. 13 & 14). Of course, also for increasing complex DNNs, the differences of representations learned using direct optimization, on the one hand, to representations of more conventionally trained VAEs, on the other, will remain very interesting. Thanks!

---

> > ### Comment · Reviewer_DBE8 · 2021-11-24
> > **I keep my score unchanged**
> >
> > Dear authors,
> >
> > I highly appreciate your hard work and the rebuttal. You did an amazing job! However, I still think the paper lacks some important comparisons and, as mentioned by other reviewers, it is hard to see whether the benefits of the proposed approach compensate the additional, rather large computational burden. Probably, this paper could get a spotlight at other conferences, e.g., GECCO or PPSN, but I doubt it is convincing enough for the audience of ICLR. Therefore, I regret to say I keep my original score.
> >
> > All the best!

---

### Official Review · Reviewer_ALx1 · 2021-10-30

**Correctness:** 4
**Technical Novelty And Significance:** 4
**Empirical Novelty And Significance:** 4
**Recommendation:** 8
**Confidence:** 4

**Main Review:**

**Strenghts**

I found the paper to be a very interesting read, since it presents a novel idea to train discrete VAEs, which is notoriously challenging. The idea is simple and theoretically grounded, as it is based on the truncated variational EM. Overall the paper is well written and clearly explains the presented method. The introduction is however too long and not to the point: most of it can be put in a "related work" section, and the introduction could be used instead to better introduce and justify TVAEs.

Discrete VAEs are commonly trained with continuous relaxations of latent variables such as using the Gumbel-Softmax distribution. This is however hard to train due to the needed annealing that is difficult to tune, and as discussed in this paper does not offer very sparse representations. The idea presented in this paper is quite different from standard VAE training, since it relies on non-amortized methods based on truncated approximations and evolutionary algorithms. This implies that this method is able to avoid the amortization gap (but also that its training will be slower and less scalable). TVAEs show SOTA/competitive performances in standard benchmarks for image inpainting and denoising, even in the one-shot setting (while other methods might require large amount of data to achieve similar results).

**Weaknesses**

The paper offers many details on the proposed algorithm both in the main text and in the very extensive appendix. However, after reading both of them, I still had many open questions on why and how this method works, which require some more detailed analyses and ablation studies. Some of them might be at least inferred by reading the appendix in detail, but due to their importance this should not be required from the reader.

The core idea of the algorithm is based on a sample-based approximation of a high dimensional integral in the ELBO, where samples are obtained with evolutionary strategies. This raises several questions:
1. How good and robust is the approximation obtained by evolutionary strategies?
    1. How would a random sample selection perform?
    2. Have you experimented with different EAs?
    3. How does the quality of this approximation varies if we change the number of latent dimensions?
2. How do you initialize the S latent states in algorithm 1? It is a bit unclear if you use the samples from the previous iteration by keeping population in memory (at the end of page 5 you say it's not required, but in page 6 you say they need to be remembered across iterations).
    1. in case you re-use samples from the previous iterations, how do you ensure you do not get such in local optima? Especially at the beginning of training the latent space will be constantly changing.
    2. How diverse are the states at the end of the evolutionary loop?

Due to the non-amortization and the usage of EA, for scalability reasons this method can only be applied to a restricted task domain, that is however well defined and discussed in the paper.

Minor comment: figures in main text are way too small to appreciate the differences, which are key in denoising/inpanting tasks.


**Summary Of The Paper:**

This work focuses on VAE models with discrete latent variables, that are optimized with evolutionary algorithms using ideas from truncated Variational Expectation Maximization. This model is shown to learn a sparser representation than other methods for discrete VAEs. This allows TVAEs to be well suited for denoising and inpainting tasks.


**Summary Of The Review:**

The presented idea is interesting and novel in the VAE setting, and could inspire new research in generative models with discrete latent variables.
While the experiments show that the model performs well in inpainting and denoising tasks, I feel that to be more impactful this paper lacks a clearer analysis and discussion on what makes this model work. I'd be happy to increase my score if the authors improved the paper in this direction.

---

> ### Author Response · Authors · 2021-11-23
> **Reply to points raised**
>
> Point 1: Yes, we experimented with different EAs (Point 1.2). Different evolutionary operators do result in different learning behavior, and in different performance. The most systematic investigation we did on artificial data (new Fig. 11). The combination “fitparents-randflips” offered the best trade-off between effectiveness and efficiency (crossover operators are computationally a bit more demanding). It was also observed to perform well on non-artifical data. Just random variation (Point 1.1) did not perform well (e.g. Fig. 11). For instance for the data enhancement applications, increasing the latent dimension $H$ in general results in better performance. But using large H is also computationally more demanding (the DNN gets larger and the posterior is higher dimensional, with the latter demanding larger $S$ to be appropriately represented).
>
> Point 2: We initialize the $\Phi^{(n)}$ by drawing from a Bernoulli distribution with $\frac{1}{H}$ probability for each $z_h$ (we added a clarification to App. B.1). Regarding the states in $\Phi^{(n)}$ (Point 2.1), they are always kept in memory. On p.5 we had the parallelization across compute nodes in mind. Neither the whole dataset nor all $\Phi^{(n)}$ have to be kept in the memory by a single node (but can be distributed across nodes). But that was ambiguous, sorry, we now changed that passage on p.5 to clarify. Regarding Point 2.2, the larger the $\Phi^{(n)}$, the more posterior structure (sparsity, correlations, modes, …) can potentially be captured. In experiments we do observe that increasing $S=|\Phi^{(n)}|$ during learning does result in higher final ELBO values*, which is evidence for improved modeling of posterior structure. If the diversity of states in $\Phi^{(n)}$would not significantly increase with larger $S$, we would not expect the ELBO values to change significantly.
>
> *For comparability: if comparing the effect of different $S$ on final ELBO values, we do have to train with different $S$, but have to compare final ELBO values using equal $S$, of course.

---

> > ### Comment · Reviewer_ALx1 · 2021-11-27
> > **Score increased**
> >
> > Thank you for the rebuttal and the updated paper. Many of my concerns have been addressed/clarified, so I have increased my score.

---

### Official Review · Reviewer_TZnV · 2021-11-01

**Correctness:** 4
**Technical Novelty And Significance:** 3
**Empirical Novelty And Significance:** 3
**Recommendation:** 8
**Confidence:** 3

**Main Review:**

This paper sheds light on a new direction of discrete latent distribution VAE models as it uses a direct model for the encoder part instead of DNN. The formula variational optimization for the direct model are novel and concrete. The criterion for encoder optimization (Eq. 10) encourages the latent code to be sparse. This is also validated in the experiments, which provides great advantages in some application scenarios. The experiment part of the paper is solid and the proposed method is consistently evaluated with previous methods. The discussion part also gives great insights into the advantages of the proposed method under different application settings.

A minor concern is the scalability of the proposed method. As pointed out by the authors, for each epoch of N data points, we need to evaluate S*N states, where S scales w.r.t. the complexity of the encoder, compared to a O(N) complexity under traditional VAE settings. This should not be a problem for a small to medium dataset but might be a performance concern for a very large dataset.

**Summary Of The Paper:**

In this paper, a novel variational autoencoder (VAE) model with binary latent distribution is proposed. Compared to traditional VAE models, aside from the latent distribution here is Bernoulli (instead of Gaussian), the encoder part is an explicitly expressed model (instead of a DNN) and can be optimized directly, which makes the model to have a simpler form compared to previous DNN-based sparse latent code learning methods. The model is numerically evaluated on (zero-shot) denoising and inpainting tasks, and shows comparable or better performance than competitive methods.

The main novelty of this paper comes from developing a direct VAE framework to handle discrete latent codes. Previous discrete latent code VAE models use DNN as encoders, and thus need methodologies, e.g., a softening of discrete distributions, for discrete latent to estimate gradients for the encoder. In this paper, the authors avoid using DNNs and directly model the encoder explicitly, whose optimization does not require gradients.

**Summary Of The Review:**

The proposed method is novel and well-presented. I would recommend accept this paper.

---

> ### Author Response · Authors · 2021-11-23
> **Importance of insights on learned representations**
>
> We thank the reviewer for the review. Datasets with large N and VAEs with large DNNs are currently a challenge for the direct approach we studied here (which we explicitly pointed out, also see new Tab. 3 added for another reviewer). Our aim is, though, not the suggestion of a novel VAE to improve task performance but a novel approach to study what type of representations have advantages and disadvantages for different types of data. And we thank the reviewer for sharing the view. The suitability of a sparse encoding for image patches can, of course, be taken as a motivation for the development of future approaches that are able to learn sparse representations also for very large scales in order to improve performance on a given task. Like the reviewer, we do, however, very much see the insights gained using the approach to be more relevant than performance increases on specific tasks. Of course, our reported results in zero-shot settings are an important validation and confirmation of the approach. But we regard the ability to gain insights on the learned representations themselves as at least as important. Thanks.

---

### Official Review · Reviewer_XRYc · 2021-11-03

**Correctness:** 3
**Technical Novelty And Significance:** 3
**Empirical Novelty And Significance:** 2
**Recommendation:** 6
**Confidence:** 2

**Main Review:**

Pros: The theoretical derivation of this article is very clear
Cons:
(1). This paper used discrete latents to solve present and absent problems of objects or edges. It would be better to prove the proposed method on a large scale dataset, such as ImageNet;
(2). The advantages mentioned by this paper are: fewer algorithm elements, fewer hyperparameters and fewer model parameters. It would be better to carry out experiments to prove that images generated from this model has better performance while maintaining fewer model parameters.


**Summary Of The Paper:**

This paper investigates VAEs with binary priors able to learn sparse latent code and studied how such codes can efficiently be learned. It also proposes a method that uses a direct discrete optimization of binary latent vectors.


**Summary Of The Review:**

This article is written clearly, but the experimental part is a bit lacking in persuasiveness.

---

> ### Author Response · Authors · 2021-11-23
> **Reply to points raised**
>
> We thank the reviewer for the points raised. Please note that a main focus of our approach is the study of representations and their suitability for different types of data. A main result is that sparse codes do seem advantageous for image patch data, while dense codes seem more advantageous for large image data sets (we used CIFAR-10). How well sparse codes vs. dense codes can perform on large image data sets such as ImageNet currently remains an open question, we would argue. Presumably larger and more intricate DNNs are required in conjunction with improved efficiency of a direct optimization (also see replies to DBE8, hasK and general).
>
> Regarding the number of parameters, note that in Fig. 15 we deliberately included the same decoder architectures for comparability. TVAE then uses no DNN parameters for the encoder but hyperparameters for the evolutionary optimization instead (DNNs have more parameters but comparison is non-trivial, of course).

---

### Official Review · Reviewer_hasK · 2021-11-07

**Correctness:** 4
**Technical Novelty And Significance:** 3
**Empirical Novelty And Significance:** 3
**Recommendation:** 5
**Confidence:** 4

**Main Review:**

I am somewhat split on this paper. On the one hand, I found the paper to be well motivated at a high level as a way to avoid gradient bias and obtain sparsity; and the approach taken by the authors is quite original and deviates from most VAE works. On the other hand, the proposed method is much more computationally intensive than regular training of VAEs, and I also wonder if this might be a case of over-complicating things for the sake of the idea sounding interesting.

The paper is mostly well-written, although I would appreciate some additional background on genetic/evolutionary optimization algorithms in the appendix, as I do not believe these are often in the area of expertise of the intended machine learning audience of this paper.

As for the experiments, while the tasks selected by the authors do highlight some benefits of their proposed method; it does not seem like the proposed approach necessarily performs better (e.g. Figure 12 in the appendix shows the proposed method achieving worse ELBO than a Gumbel-Softmax VAE with a comparable architecture, although the proposed method recovers sparser latents). I believe there should be a discussion about these results in the main manuscript. Finally, the authors do discuss computational cost, but do not report running times. Given that the cost is highly increased over the baselines, I believe it's particularly important for the authors to report running times. Finally, the only continuous relaxation the authors compare against is the Gumbel-Softmax, and not further improved versions, e.g. [1,2,3].

Minor things:

-abstract: "canonically" -> "canonical"

-change "a.k.a." to something more formal, e.g. "i.e."

-page 3, second paragraph: "Gumble" -> "Gumbel"



[1] Estimating Gradients for Discrete Random Variables by Sampling Without Replacement, Kool et al., ICLR 2020

[2] Invertible Gaussian Reparameterization: Revisiting the Gumbel-Softmax, Potapczyski et al., NeurIPS 2020

[3] Rao-Blackwellizing the Straight-Through Gumbel-Softmax Gradient Estimator, Paulus et al., ICLR 2021

========================================================================================================

UPDATE 1 AFTER REBUTTAL

========================================================================================================

I have read the other reviews, as well as the author's rebuttal, and unfortunately my concern that this paper is complicated for the sake of it remains. I will thus maintain my current score.

**Summary Of The Paper:**

This paper proposes an approach to learn VAEs with binary latent variables. Rather than relying on commonly used continuous relaxation techniques, the authors completely forgo the encoder neural network and parameterize the approximate posterior using a subset of states, so that the approximate posterior is proportional to the true posterior but only supported in the corresponding subset of states. The authors then propose a training algorithm which involves doing gradient descent over the decoder and using a genetic algorithm to optimize over the (non-amortized) posterior distribution.

**Summary Of The Review:**

This paper proposes an interesting and novel way of training VAEs with binary latents; although the method adds a lot of computational complexity for benefits which are not completely clear.

---

> ### Author Response · Authors · 2021-11-23
> **Reply to points raised**
>
> We followed the suggestion and added a comparison of wall-clock runtimes for GSVAE, VLAE and TVAE. See new Tab. 3. For the example of the 'Barbara' image, VLAE and TVAE both require a few hours for denoising (6.9 and 9.5, respectively). GSVAE is much faster with just a few minutes. The new Tab. 3 now simply quantifies what was explicitly stated before: TVAE is computationally demanding but at scales at which it can be applied more effectively.
>
> Regarding how the studied approach may be evaluated, please see our general reply to all reviewers. We add to the general reply the following:
>
> If considered (Point B of general reply) merely as a method to improve performance, then we kindly refer to the denoising and inpainting benchmarks of Sec. 3 (plus Appendix). Given the ‘zero-shot’ setting of just one noisy image, the task is to compute an image with as much noise removed as possible. In other tasks with more images, a method with higher efficiency usually means that larger DNNs can be trained. And larger DNNs in turn usually mean better performance. The same is not true in the ‘zero-shot’ setting because limited data means only DNNs of limited size can be trained (and longer training e.g. of GSVAE also does not improve performance).
>
> If considered (Point A of general reply) as a tool to gain insights into suitable representations for image patches, then note the qualitative observation we report. Concretely, note the observation that sparse codes seem favorable, and that standard VAE approaches (including discrete VAEs) favor dense codes. The insights go hand-in-hand with the method itself: only with the very different optimization we study, the qualitatively very different representations can be investigated. One direct consequence of our results is, e.g., an encouragement of research on representations with sparse codes for data enhancement tasks.
>
> Coming back to the inquired computational efficiency, do note that wall-clock runtimes reported by our new Tab. 3 are not necessarily a very objective measure. Prototype source code using non-mainstream approaches (such as VLAE or TVAE) is typically not optimized very much. While it is difficult for us to say much about the potential to further increase VLAE efficiency, the potential to further increase TVAE certainly lies in improved IO management of memory to CPU information transfer potentially in combination with flexible $\Phi^{(n)}$ sizes, use of improved data-structures like soft-heaps and so forth. GSVAE, on the other hand, has relatively conventional algorithmic components that can all be implemented as standard toolbox routines (e.g. in TensorFlow or PyTorch), and such routines are usually highly optimized for efficiency.
>
> Further to (B): We simply chose the most standard and most widely used baseline (GSVAE) for comparison. TVAE’s mainly differs from GSVAE by (1st) using a very flexible posterior representation (i.e., sets of states), while GSVAE does assume a fixed categorical distribution as posterior model; (2nd) TVAE is non-amortized, while GSVAE uses an encoder DNN; and (3rd) TVAE uses direct discrete optimization where GSVAE uses sampling-based estimations of a gradient. These differences do distinguish TVAE from GSVAE as well as from all three references [1-3] suggested (which we now discuss, and thank the reviewer for pointing them out). Furthermore, [1-3] like GSVAE cannot directly be applied to inpainting.
>
> Regarding (former) Fig. 12, sure, advantages in task performance are not observed for all tasks, as we explicitly pointed out. And sparse codes are not always preferable to dense codes, true. We’ve stressed this in the manuscript and made changes to work this out more clearly. We do, in general, think that knowing and discussing the advantages and disadvantages of an approach helps to better place the approach into context.

---

### Author Response · Authors · 2021-11-23
**General reply to all reviewers**

We thank the reviewers for appreciating the novelty of the approach, and its properties. We also thank the reviewers for the many pointers to additional relevant papers. Our paper is focused on an approach diverting significantly from the usual approach. An approach with such significant differences has, of course, advantages and disadvantages. While our paper is very explicit about disadvantages (such as computational demand), we describe two main advantages: (A) the novel approach allows for studying different representations (sparse vs. non-sparse) and their suitability for different data (image patches, images of whole objects, we also added acoustic data); and (B) the novel approach results in competitive denoising and competitive inpainting performance for the ‘zero-shot’ setting.

Point (A), i.e. the possibility to study novel representations for prominent data, we regard as at least as relevant as point (B). We kindly refer to the last two sentences of our abstract, for instance. Competitive performance in the ‘zero-shot’ setting, notably for denoising as well as inpainting, can be regarded as a significant contribution in its own right, we agree. But we hope that our manuscript does not make the impression of a specific task being the selling point of the approach. The additionally suggested papers (reviewers “hasK” and “DBE8”) share with GSVAE and TVAE that encoder DNNs are optimized using (estimates of) gradients. We fully agree that alternatives/improvements for that class of approaches is important to improve performance or black-box applicability (we now discuss the suggested papers in Sec. 1). Of course, we also can not rule out for sure that any of the suggested approaches may achieve SOTA results for ‘zero-shot’ tasks (although the similarity of most approaches to GSVAE, which we used as accepted baseline, does not suggest a necessarily strong performance). However, instead of performance (Point B above), the study of the type of representation and their suitability e.g. for image patches (Point A) is much less the focus of any of the suggested approaches. And we here provide evidence for amortization and the specific choice of a posterior model (without correlations or with fixed a categorical distribution to model posteriors as for GSVAE) has a strong effect on the learned representation (i.e. dense rather than sparse).

We have rephrased some text to work out this point more clearly. Furthermore, the introduction now contains a separate section to discuss related work (→ Rev. ALx1), and we included further approaches for gradient optimization with discrete random variables (→ Rev. DBE8) and further improved Gumbel-softmax versions (→ Rev. hasK).
In Sec. 2, we now put more emphasis on the non-gradient based encoder optimization to improve the introduction of the key elements of TVAE (→ Rev. ALx1). Secs. 2, 3 and App. B.1 are now also more explicit about the decoder networks used (→ Rev. DBE8); also see the new Tab. 1 which lists all hyperparameters used in the numerical experiments.

We also made further changes following the suggestions in the reviews. Regarding points relevant for more than one of the five reviews, we did the following:

In Appendix A.1 and Fig. 11, we now provide additional details about the evolutionary optimization, discuss different genetic operator combinations, and motivate the ones used in the numerical experiments (→ Revs. DBE8, ALx1, hasK); we also added a comparison with random sampling (see Fig. 11).
App. B.4 (Comparison to Deep Generative Models paragraph) and B.6 now provide further details about our comparisons with gradient-based encoding approaches, including Gumbel-softmax (→ Revs. DBE8, hasK). We now also report denoising performance of GSVAE for House in Fig. 3.

Lastly, we added the new Appendix B.7 with new numerical results for a ‘zero-shot’ audio denoising benchmark, and we have uploaded audio files belonging to this benchmark as part of the supplementary material.

Regarding points raised in just one review, please see our replies to the individual reviews.

---

### Public Comment · ~Jorg_Lucke1 · 2023-04-28
**Paper Accepted at Next Major Conference**

We submitted our paper to the next major Machine Learning conference (which was ECML 2022), where it was accepted after three positive reviews. The final version of the paper is since March 2023, available here:

https://doi.org/10.1007/978-3-031-26409-2_22

To the reader of this public OpenReview page:

Please note that the public ICLR meta-review below is not reflecting the positive ICLR reviews and the positive and constructive rebuttal of our ICLR 2022 submission (we received final scores of 8, 8, 6, 5, 5). We can not say why the text of the meta-review was then so negative, and why it contained statements unsupported by the in-depth reviews and rebuttal (we elaborated in a previous response further below). Our non-public request to change the ICLR meta-review text accordingly remained unanswered.

---

### Decision · Program_Chairs · 2022-01-20

**Decision:**

Reject

**Comment:**

This paper presents a new approach for learning binary latent variable models using evolutionary optimization.

Pros:
* A new perspective to learning binary latent variables is proposed using evolutionary algorithms.
* The proposed method works well on auxiliary tasks such as zero-shot denoising and inpainting.

Cons:
* The proposed evolutionary optimization performs poorly on the binary VAE problem.
* It has a high computational cost that will limit its application in real-world problems.
* An in-depth comparison with prior work on learning discrete latent variables is missing. This may include MCMC-based approaches, REINFORCE-based techniques, REBAR or RELAX.

This paper presents an interesting direction for learning binary latent variables using evolutionary algorithms. However, the proposed method performs poorly on the binary VAE problem which is the core problem, targetted in this paper (See likelihood values for binary VAEs in Fig. 15). The reviewers have raised concerns regarding the computational complexity of the evolutionary method in practice. They have also criticized the missing baselines for the binary VAE experiments.

The authors have argued that the proposed method excels at auxiliary problems such as zero-shot image denoising and inpainting. However, these problems are not the central problem of this submission, and naturally, they have not been discussed, reviewed, and evaluated thoroughly. They can be also addressed with non-binary VAEs and other forms of generative models which are not discussed in the paper.

Given these concerns, we don't believe that this submission in its current form is ready for publication at ICLR.

---

> ### Public Comment · ~Jorg_Lucke1 · 2022-04-08
> **Regarding mismatch of meta-review text and reviews/rebuttal**
>
> The paper received reviewer scores of 8, 8, 6, 5, 5. We knew that the final 6.4 average was a borderline case for a competitive venue, so a rejection is common business. Still, we were disappointed about the reject decision.
>
> We would leave it as is but now OpenReview makes the meta-review (and author names) public, it seems. While accepting the decision itself, the decision *text* now publicly gives an impression that does not reflect the positive and generally supportive review process (and the good scores). Even the most negative (i.e. "5") reviews provide (we would very much say) a more balanced view than the meta-review.
>
> Most importantly regarding scientific content: There is no such thing as "the binary VAE problem", and none of the reviewers claimed so. A nice asset of our field is that we investigate a broad diversity of approaches for a broad diversity of tasks. Some approaches work well on some tasks, others well on other tasks. It is considered good scientific practice for researchers to not only point to tasks and settings on which their approach works well but to also point out limitations and tasks where an approach does not work well. This is what we did when we documented the approach's competitive performance on image patch data, while we also reported non-SOTA performance on datasets of large, single-object images (we used CIFAR as example). If the meta-review now states (as basically main rejection reason) that the approach "performs poorly on the binary VAE problem", you make it sound as if you believe that one specific task is of interest only - and that all others are "auxiliary tasks". (A) We don't believe that this was intended, (B) denoising and inpainting with many papers each year are no auxiliary tasks, and (C) also for the non-SOTA task no reviewer wrote "poor performing" (please see review details; Fig. 15 provides a comparison based on the constraint of elementary DNNs & binary latents vs. continuous latents).
>
> We have therefore asked to change the public meta-review accordingly, also in the interest of future ICLR authors who may otherwise feel discouraged to also report non-SOTA performance on some tasks. We also asked to amend "missing (...) in-depth comparison with prior work". We do provide an in-depth comparison (see our paper) while providing an exhaustive list of references is always difficult, of course. The third "con" point of a high computational demand is valid, and it was explicitly discussed in paper, reviews and rebuttal.
>
> We posted the request to change the meta-review non-publically five weeks ago.
>
> As we haven't received any response and as the meta-review is public, we now post this response publicly.
>
>
>
> P.S.: While synchronized with all authors, the text itself was written by the senior author.